# CHAIN-OF-THOUGHT PROVABLY ENABLES LEARNING THE (OTHERWISE) UNLEARNABLE

**Chenxiao Yang**[†,*], **Zhiyuan Li**[†], **David Wipf**[§]
† Toyota Technological Institute at Chicago, § Amazon Web Services
{chenxiao,zhiyuanli}@ttic.edu, davidwipf@gmail.com

## ABSTRACT

Modern language models have demonstrated remarkable reasoning capabilities by using chain-of-thought (CoT). One hypothesis about the inner workings of CoT is that it breaks down originally complex tasks into smaller subtasks that are more amenable to learning. We formalize this by showing possibility and impossibility results of learning from in-context demonstrations with and without CoT. In particular, with CoT, we examine a family of learning algorithms that learn a task step-by-step, capable of composing simpler functions from individual reasoning steps to form an overall complex function. This process reduces the difficulty of learning a task to that of the hardest reasoning step in the chain. Moreover, we prove Transformers can express this algorithm and thus they can efficiently in-context learn arbitrary tasks as long as these tasks can be decomposed into a finite number of subtasks, each of which are efficiently learnable. In contrast, without CoT, we demonstrate that there exist tasks that are inherently unlearnable by the same algorithm. Overall, our results suggest several provably effective ways for decomposing target problems to instantiate CoT. Empirically, we demonstrate our proposed CoT construction significantly enhances the reasoning capabilities of real-world LLMs in solving challenging arithmetic reasoning tasks, including learning polynomials and Boolean formulas.

## 1 INTRODUCTION

Complex problem solving often involves breaking down an originally challenging task into smaller, more manageable subtasks, learning from these subtasks, and then composing the acquired skills to address the overall task — a strategy that reflects how humans naturally solve problems. One empirically-successful method that mimics this process is called Chain-of-Thought (CoT) (Wei et al., 2022; Reynolds & McDonell, 2021; Nye et al., 2021), whereby a model is provided with demonstrations involving detailed reasoning steps and subsequently instructed to generate thoughts step-by-step before yielding the final answer. Modern language models rely on CoT or variants thereof (Yao et al., 2024; Besta et al., 2024) to tackle complex tasks ranging from commonsense reasoning to mathematical proofs (Cobbe et al., 2021; Rae et al., 2021; Srivastava et al., 2022), sometimes even exceeding the capabilities of human experts.

Despite the empirical success of CoT, theoretical investigation thus far has remained relatively sparse. Some previous efforts have made strides from the perspective of Transformer expressiveness (Li et al., 2024; Feng et al., 2024) or through case studies of in-context learning MLPs (Li et al., 2023b). However, these works either focus on mostly generic cases without specifying concrete/actionable ways of actually decomposing a task, or involve quite restricted scenarios where both the task and the CoT are predetermined. Since in practice not all intermediate steps in CoT are equally useful, e.g. Zhang et al. (2022); Press et al. (2022), and the target tasks are typically diverse, there still exists a considerable gap between current theory and practical scenarios. In targeting this gap, we ask the following core question:

*How do task decompositions affect the ability of a model to learn complex reasoning tasks, and what principles can guide the corresponding optimal CoT design?*

---

*Work was done during author's internship at Amazon Web Services.

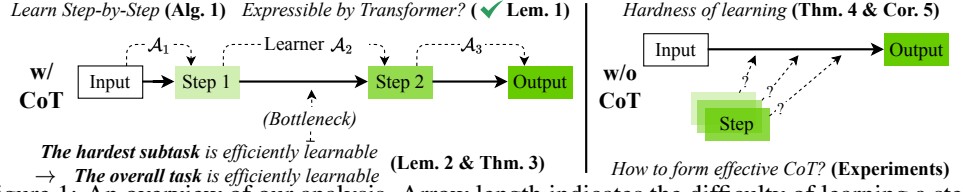

Figure 1: An overview of our analysis. Arrow length indicates the difficulty of learning a step.

One possible explanation for the efficacy of CoT is that it reduces the difficulty of learning a complex task to the level of learning a series of subtasks. The implication here is simply that, assuming these subtasks are suitably designed and orchestrated, then the necessary skills acquired at the subtask level through CoT will suffice to outperform attempts at learning directly from the overall task. In this work, we formalize these intuitions in a mathematically rigorous way, and take initial steps towards quantifying the benefits of different task decomposition schemes.

**Problem Setup.** To define a notion of learnability in the context of language modeling, we adopt the classic PAC learnability (Valiant, 1984) in the setting where the learner is associated with a parametric model — a setting known as in-context learning (Brown et al., 2020; Garg et al., 2022). Unlike conventional supervised learning, the learning considered here occurs during test time, which a highly useful feature enabling the model to adapt to new tasks that were not explicitly seen in pre-training. Concretely, we say a task is in-context learnable by a parametric model $\mathcal{M}$ if there exists a parameter configuration $\theta$ such that, for arbitrary distribution from the task, the model can, upon receiving a set of i.i.d. demonstrations/samples $D$ and a query/test sample $x$, output a prediction for the query, i.e. $\mathcal{M}_\theta(D, x)$, that is close to the ground-truth $y$ with high probability. A learning algorithm $\mathcal{A}$ is implicit in this process, which takes $D$ as input and outputs a predictor or hypothesis $h_D : \mathcal{X} \to \mathcal{Y}$. The relation between the algorithm and the model $\mathcal{M}_\theta$ can be written as

$$\mathcal{M}_\theta(D, x) = \mathcal{A}(D)(x) = h_D(x). \tag{1}$$

To account for the effects of CoT, we also formalize the notion of 'task decomposition' as transforming the distribution from which $D$ is generated into a sequence of distributions that generate $D$ with detailed intermediate steps. (Section 2)

## 1.1 CONTRIBUTIONS

**Main Results.** To formally answer whether or not CoT can help a model learn complex reasoning tasks, we investigate the in-context learnability of tasks w.r.t. different decomposition schemes. We find the answer is often affirmative depending on the task decomposition and summarize our findings via the following two informal statements:

1. Regardless of how complex a reasoning task is, it can be efficiently learned by Transformers in-context as long as it can be decomposed into a finite number of reasoning steps, each of which is efficiently learnable by a learner Transformers can perform. (Section 3)

2. There exist inherently hard tasks that are not learnable without CoT regardless of the sample size, and yet nonetheless become learnable by using CoT with specific decomposition schemes we introduce. (Section 4)

In aggregate, our results suggest a broad range of scenarios where CoT can indeed effectively reduce the hardness of learning, from that of the overall task to that of the hardest reasoning step in the chain, or even from an unlearnable level to the learnable level. These results also suggest several actionable ways to form effective intermediate steps of CoT.

To obtain the above results, we introduce a class of learning algorithms $\mathcal{A}_{\text{CoT}}$ enabled by CoT, dubbed step-by-step learning. This class of algorithms takes as input CoT examples, and outputs a complex predictor $h_D$ by composing predictors $\{h_i\}_{i \in [k]}$ obtained from $k$ individual algorithms, where $k$ is the number of reasoning steps (Algorithm 1). Given a fixed $k$, the expected overall prediction error made by $h_D$ can be upper bounded by the individual errors made by predictors $\{h_i\}_{i \in [k]}$ on their respective reasoning steps (Lemma 2). Leveraging this fact, we prove that the difficulty of learning the overall task can be reduced to that of the hardest constituent step of CoT, since the sample complexity of this step determines the sample complexity of learning the overall task (Theorem 3).

Furthermore, we establish that the capabilities of $\mathcal{A}_{\text{CoT}}$ described above can be achieved by Transformers — the de facto parametric model $\mathcal{M}$ used in language modeling. Specifically, we show

that a linear depth (w.r.t. $k$) and constant embedding size Transformer is sufficient to express $\mathcal{A}_{\text{CoT}}$ in an end-to-end manner, if the individual algorithms for learning each step are instantiated as kernel gradient descent that produces a predictor $h_i : x \mapsto W\phi(x)$ with a non-linear feature map $\phi(\cdot)$ (Lemma 1). Combining these results suggests that a reasoning task is efficiently learnable by Transformers in-context as long as each subtask is efficiently learnable by these individual algorithms.

Within the same analytical framework, we further present impossibility results illustrating the hardness of learning without CoT. Particularly, we consider the task of learning a Boolean function class called sparse parity. Without CoT, this task is impossible to learn by the variant of algorithm $\mathcal{A}_{\text{CoT}}$ with $k = 1$ due to limited approximation power of the hypothesis class that is used to learn a single step, whereas if we enable CoT with specific intermediate reasoning steps, the task becomes learnable with guarantees of smaller errors (Theorem 4 & Corallary 5).

**Experiments.** For empirical verification, we consider new arithmetic reasoning tasks and test if the decomposition schemes from our analysis are practically effective (Section 5). Specifically, we propose to construct complex reasoning tasks with varying overall hardness and hardness of subtasks. We observe that the performance of real-world LLMs improve significantly as the difficulty of the hardest step in the CoT reduces, regardless of the overall task complexity. Additionally, on learning two inherently hard Boolean functions, we demonstrate that our introduced CoT significantly enhances reasoning performance, sometimes improving accuracy from nearly random guessing to nearly perfect.

## 1.2 RELATED WORK

CoT has been analyzed through the lens of the expressiveness of finite-precision Transformers Feng et al. (2024); Li et al. (2024) . In particular, Feng et al. (2024) proves CoT can enable Transformers to solve some specific tasks such as basic arithmetic and linear equations, while Li et al. (2024) proves that increasing the step number in the CoT allows them to emulate circuits of increasing depth. More closely related to ours, Li et al. (2023b) studies a scenario in which the intermediate steps of CoT are intermediate layers of an MLP, showing that Transformers can in-context learn the MLP. We also note that there are many recent papers that empirically analyze CoT, e.g. (Wang et al., 2022; Fu et al., 2023b; Madaan & Yazdanbakhsh, 2022; Turpin et al., 2024; Prabhakar et al., 2024) among others. **However, previous works are generally limited in one of two ways: 1)** either the problem setting is overly restricted, focusing on fixed tasks and specific forms of decomposition, such as solving arithmetic equations (Feng et al., 2024) or learning MLPs (Li et al., 2023b), **2)** or in a more complex regime (Li et al., 2024), where though the possibility for improvement is shown, the effects of specific intermediate steps are not quantified. In contrast, our work introduces a new setting, one which is general enough to encompass learning all distribution families, while also explicitly accounting for the effects of specific decomposition schemes. Our contribution is also related to a recent line of work on ICL (Akyürek et al., 2023; Von Oswald et al., 2023; Dai et al., 2023; von Oswald et al., 2023; Cheng et al., 2024; Li et al., 2023a), particularly demonstrations of how Transformers with certain weight constructions perform ICL similarly to optimization algorithms like gradient descent (Von Oswald et al., 2023) or its variants (Giannou et al., 2024; Fu et al., 2023a). Notably, Cheng et al. (2024) proves that Transformers could implement kernel gradient descent, which is closely related to ours. The idea of task decomposition and generalizing from easier to harder tasks has also been explored in other works of learning theory, e.g. Natarajan (1989); Tadepalli (2008); Wies et al. (2022).

## 2 PRELIMINARIES

**Chain-of-Thought (CoT).** We follow the commonly-adopted few-shot CoT setting, where demonstrations are augmented with intermediate steps and the prediction is also in the format of CoT. In particular, for $k$ reasoning steps, we denote each demonstration as

$$e = (x, z_1, \cdots, z_{k-1}, y) \in \mathcal{X} \times \mathcal{Z}_1 \cdots \mathcal{Z}_{k-1} \times \mathcal{Y} \tag{2}$$

where $z_t \in \mathcal{Z}_t$ represents the $t$-th intermediate reasoning step. For convenience, we also let $z_0 = x$ and $z_k = y$. Let $d(\mathcal{Z}_i)$ be the dimension of $\mathcal{Z}_i$, and $d = \sum_{i=0}^{k} d(\mathcal{Z}_i)$.

**In-Context Learning (ICL).** In ICL, the base model is provided with $N$ demonstrations or in-context examples $e^{(i)} = (x^{(i)}, y^{(i)})$ for $i \in [N]$ where $x \in \mathcal{X}$ and $y = f(x) \in \mathcal{Y}$. We denote a learning algorithm or learner as $\mathcal{A} : (\mathcal{X} \times \mathcal{Y})^N \to \mathcal{H}$, which takes a set of demonstrations

as input and outputs a predictor or hypothesis $h : \mathcal{X} \to \mathcal{Y}$ from a hypothesis class $\mathcal{H}$. Given a set of demonstrations and a query $x^{(N+1)}$ (which we assume is from the same distribution as the demonstrations), the goal of the base model is to learn a predictor $h$ and use this predictor to make predictions on the query $x^{(N+1)}$. For base model Transformer $\mathrm{TF}_\theta(\cdot)$ with parameters $\theta$, we have

$$\mathrm{TF}_\theta(\{(x^{(i)}, y^{(i)}) : i \in [N]\}, x^{(N+1)}) = \mathcal{A}(\{(x^{(i)}, y^{(i)}) : i \in [N]\})(x^{(N+1)}) = h(x^{(N+1)}). \quad (3)$$

**Transformers.** Let $E = \{e^{(i)} : i \in [N]\} \in \mathbb{R}^{d \times N}$ be the concatenation of samples, and let $e^{(N+1)} = (x^{(N+1)}, 0) \in \mathbb{R}^d$ whose dimension aligns with other examples. Following prior work (Von Oswald et al., 2023; Ahn et al., 2023; Cheng et al., 2024; Zhang et al., 2023), a single-head self-attention layer with weights $W_K, W_Q, W_V \in \theta$ updates $e^{(N+1)}$ as

$$e^{(N+1)} \leftarrow e^{(N+1)} + W_V E \sigma(E^\top W_K^\top W_Q e^{(N+1)}), \quad (4)$$

where $\sigma$ is non-linearity that could be specified as Softmax, ReLU or some kernel functions, e.g. (Choromanski et al., 2021; Katharopoulos et al., 2020; Wang et al., 2020; Peng et al., 2021). Stacking multiple self-attention layers (with or without an MLP module applied between each self-attention layer) gives us the Transformer considered in this paper. Following (Von Oswald et al., 2023), and for subtle technical reasons related to the construction in Sec 3.2, we exclude the query token when computing the attention.

**Our Setup.** We are now ready to formally define the problem setup. Let $\mathcal{D}$ be a distribution over $\mathcal{X}$, and $f : \mathcal{X} \to \mathcal{Y}$ a target function. An input distribution and target function pair $(f, \mathcal{D})$ defines the generating process of in-context examples, namely examples are drawn based on $x \sim \mathcal{D}, y = f(x)$. Let $\mathcal{P}$ be a family of distributions defined as a set of $(f, \mathcal{D})$ pairs, representing a certain task a model aims to solve. For instance, the target functions in $\mathcal{P}$ could be defined as all polynomials, Boolean functions, etc. The following error quantifies how successfully an algorithm can learn the task:

$$\Delta(\mathcal{P}, \mathcal{A}) \triangleq \max_{(f, \mathcal{D}) \in \mathcal{P}} \mathbb{E}_{x \sim \mathcal{D}} \left[ l\left(h(x), f(x)\right) \right] \quad (5)$$

where $l$ is the squared loss (which could be extended to other convex loss functions), $h$ is the predictor given by a learner $\mathcal{A}$ on $N$ i.i.d. examples from $(f, \mathcal{D})$. Minimizing this error guarantees successful learning of all target functions within the family $\mathcal{P}$.

**Definition 1** (In-Context Learnability). *We say a parametric model $\mathcal{M} : \theta \mapsto \mathcal{M}_\theta$ (i.e. a functional mapping from parameter space to function space) can learn task $\mathcal{P}$ if there exists a learning algorithm $\mathcal{A}$ and a function $N_\mathcal{A} : (0, 1)^2 \to \mathbb{N}$, such that for any confidence and accuracy parameters $\delta, \epsilon \in (0, 1)$:*

1. *Under a certain parameter choice $\theta$, we have $\mathcal{M}_\theta(\cdot, x) = \mathcal{A}(\cdot)(x)$ for any query $x$.*
2. *Given $N_\mathcal{A}(\delta, \epsilon)$ i.i.d. examples, the algorithm $\mathcal{A}$ returns a predictor $h$ such that with probability of at least $1 - \delta$, $\Delta(\mathcal{P}, \mathcal{A}) \leq \epsilon$.*

Moreover, we say $\mathcal{P}$ can be efficiently in-context learned by $\mathcal{M}$ if both the running time of $\mathcal{A}$ and the sample size $N_\mathcal{A}(\delta, \epsilon)$ are polynomial in $\delta^{-1}$ and $\epsilon^{-1}$. Note that the notion of learnability here is different from the classic PAC learnability (Shalev-Shwartz & Ben-David, 2014) in the sense that, per in our definition, the model itself acts not as a predictor, but a learner. This is a highly desirable property for modern language models as it allows them to meta-learn out-of-distribution tasks that were not available during pre-training (Brown et al., 2020). In the rest of this paper, we stipulate the model as a Transformer $\mathrm{TF}_\theta$ as defined via (4), unless otherwise stated.

Moving forward, we focus on whether or not CoT can improve the in-context learnability of different reasoning tasks. In principle this can be studied from multiple vantage points, such as the the effect of CoT on the sample efficiency $N_\mathcal{A}(\delta, \epsilon)$, or the existence of cases where a task is initially not learnable but becomes so once CoT is enabled, etc. Notably, addressing these issues depends critically on the specific forms of intermediate CoT steps involved. To accommodate this aspect, we next formalize the notion of a task decomposition.

**Definition 2** (Task Decomposition). *A decomposition operator $T$ is such that, for every $(f, \mathcal{D}) \in \mathcal{P}$, a target function can be decomposed as $T(f) = (f_2, f_1)$ subject to $f = f_2 \circ f_1$, where $f_1 : \mathcal{X} \to \mathcal{Z}$ and $f_2 : \mathcal{Z} \to \mathcal{Y}$ for another space $\mathcal{Z}$. This operation induces two new distribution families*

$$\mathcal{P}_1 = \{\{(f_1, \mathcal{D}) : (f_2 \circ f_1, \mathcal{D}) \in \mathcal{P}\}\} \quad and \quad \mathcal{P}_2 = \{\{(f_2, \mathcal{D}') : (f_2 \circ f_1, \mathcal{D}) \in \mathcal{P}\}\} \quad (6)$$

*where $\{\{\cdot\}\}$ is multiset allowing repeating elements, and $\mathcal{D}' : \mathcal{Z} \to \mathbb{R}$ is determined by $f_1$ and $\mathcal{D}$.*

The decomposition always exists and is not unique (e.g. $f_1$ can be arbitrary bijection). Particularly when $f_1$ is specified as the identity mapping, we have $\mathcal{P}_2 = \mathcal{P}$, in which case CoT is unlikely to work. For each decomposition, we can associate it with the generating process of demonstrations (denoted as $z$), i.e. $x \sim \mathcal{D}$, $z = f_1(x)$, $y = f_2(z)$. This affects how examples are generated in Definition 1, and thus could impact the learnability of a task. Note that this definition can be generalized to CoT with $k > 2$ steps by sequentially applying the decomposition.

# 3 Improved Learnability by Task Decomposition

To demonstrate how CoT improves learnability by task decomposition, we will first define a learning algorithm $\mathcal{A}_{\text{CoT}}$ (Sec 3.1) and prove that it can be expressed by Transformers with linear depth (Sec 3.2). Then, we derive an upper bound for the overall error, which allows us to show CoT can adjust the learnability of a complex task to the learnability of simpler subtasks (Sec 3.3).

## 3.1 Learning Algorithms Enabled by Chain-of-Thoughts

Consider a class of learning algorithms $\mathcal{A}_{\text{CoT}}$ enabled by CoT, which involves several individual algorithms $\{\mathcal{A}_i\}_{i=1}^k$ and learns increasingly complex compositional functions with more CoT steps:

---

**Algorithm 1:** Step-by-Step Learning with CoT ($\mathcal{A}_{\text{CoT}}$)

**Input:** Demonstrations $\{(z_0^{(j)}, z_1^{(j)}, \cdots, z_k^{(j)})\}_{j=1}^N$, individual learning algorithms $\{\mathcal{A}_i\}_{i=1}^k$.
**Output:** Predictor $h : \mathcal{X} \to \mathcal{Y}$
**for** $i = 1, \cdots, k$ **do**
$\quad \lfloor \ h_i \leftarrow \mathcal{A}_i(\{(z_{i-1}^{(j)}, z_i^{(j)})\}_{j=1}^N)$
$h \leftarrow h_k \circ \cdots h_2 \circ h_1$

---

This class of algorithms take as input a set of demonstrations, where each demonstration contains $k$ reasoning steps, and outputs a predictor $h : \mathcal{X} \to \mathcal{Y}$. The learning is performed in a step-by-step manner; that is, for each reasoning step $i \in [k]$, an individual algorithm $\mathcal{A}_i$ is used to learn a predictor $h_i : \mathcal{Z}_i \to \mathcal{Z}_{i-1} \in \mathcal{H}_i$. The learned predictors $h_1, h_2, \cdots, h_k$ are then composed to obtain the desired overall predictor. Note that the algorithm can be naturally extended to scenarios where each step is a function of all preceding steps, by redefining $z_i$ in the algorithm as a concatenation of $\{z_j\}_{j \leq i}$ in the initial CoT. Therefore, without loss of generality, we assume that the CoT satisfies the Markov property, meaning that each step is conditionally dependent only on the last step.

## 3.2 Expressiveness of Transformers

Next, we demonstrate parameter choices $\theta$ that connect Transformers $\text{TF}_\theta$ and algorithm $\mathcal{A}_{\text{CoT}}$ instantiated in a certain way.

**Instantiation of $\mathcal{A}_{\text{CoT}}$.** While the algorithm could have many different instantiations, in this paper, we define $\mathcal{A}_i$ as empirical risk minimization: using gradient descent to minimize a squared loss $\mathcal{L}_i$ over in-context examples to learn a predictor from a hypothesis class $\mathcal{H}_i$, which is defined as a linear model on fixed non-linear features. Specifically,

$$\mathcal{L}_i = \frac{1}{2} \sum_{j=1}^N \|h_i(z_{i-1}^{(j)}) - z_i^{(j)}\|_2^2, \quad h_i \in \mathcal{H}_i = \{z_{i-1} \mapsto W_i \phi_i(z_{i-1}) : \|W_i\|_2 \leq B\}, \quad (7)$$

where $\phi_i : \mathcal{Z}_{i-1} \to \mathbb{R}^K$ is a non-linear feature map to a $K$-dimensional space, and $W_i \in \mathbb{R}^{d(\mathcal{Z}_i) \times K}$ are learnable weights initialized to zero and subsequently with norm bounded by $B$. Therefore, the overall predictor $h$ obtained from this composition can be written as a stacked sequence of multiple non-linearities and linear transformations, i.e.

$$h = W_k \phi_k(\cdots(W_2 \phi_2(W_1 \phi_1(x)))) \in \mathcal{H} = \mathcal{H}_k \circ \cdots \circ \mathcal{H}_2 \circ \mathcal{H}_1. \quad (8)$$

For example, if $\phi_1$ is specified as the identity mapping, whereas $\phi_i$ for $i \neq 1$ are conventional activation functions, (8) could represent a $k$-layer deep neural network. And beyond this, for generic feature maps, $h$ could represent more powerful functions. As the step number $k$ increases, the predictor $h$ also becomes more complex. Below, we connect the algorithm with Transformers.

**Lemma 1** (Transformers Learn Step-by-Step). *For any $k > 1$, given a set of CoT demonstrations with $k$ reasoning steps as per (2) and a query $x^{(N+1)}$, Transformers with linear depth $kt$ and constant embedding size $2d$ can express $\mathcal{A}_{\mathrm{CoT}}$, where $\mathcal{A}_i$ is $t$ steps of GD on a squared loss and $\mathcal{H}_i$ is the hypothesis class defined in (7) whose feature map aligns with the attention as is specified in the Appendix A.1.*

*Proof Sketch.* Consider a simplified case $k = 1$. The loss is $\mathcal{L} = \sum_{i \in [N]} \|W\phi(x^{(i)}) - y^{(i)}\|_2^2 / 2$, and GD with a fixed step size updates the weights as $W \leftarrow W - \eta\nabla_W\mathcal{L}$. This process also induces dynamics in function space, i.e. the evolution of the learned predictor $h$ as the weights update. For query $x^{(N+1)}$, the function-space dynamics could be written as (see derivation in the proof)

$$\textbf{(GD Dynamics)} \quad h(x^{(N+1)}) \leftarrow \underbrace{h(x^{(N+1)})}_{\textbf{Predictions}} + \underbrace{\eta\,(Y - \hat{Y})}_{\textbf{Residuals}}\ \underbrace{\phi(X)^\top\phi(x^{(N+1)})}_{\textbf{Kernel Function}}, \tag{9}$$

where $Y = [y^{(i)}]_{i=1}^N \in \mathbb{R}^{d(\mathcal{Y})\times N}$, $\hat{Y} = [h(x^{(i)})]_{i=1}^N$, $\phi(X) = [\phi(x^{(i)})]_{i=1}^N \in \mathbb{R}^{K\times N}$. The residuals $Y - \hat{Y}$ are equivalent to $Y$ at initialization as the weights are initialized to zero. The last term represents a kernel function w.r.t. the feature map $\phi$, quantifying the similarity between the test (i.e. query) and training examples (i.e. demonstrations).

For comparison, we also rewrite the self-attention layer, where $e^{(i)}$ is reinterpreted as a concatenation of input and the residual $(x^{(i)}, y^{(i)} - h(x^{(i)}))$, which at initialization is equivalent to $(x^{(i)}, y^{(i)})$:

$$\textbf{(Transformer Layer)} \quad e^{(N+1)} \leftarrow \underbrace{e^{(N+1)}}_{\textbf{Skip Connection}} + \underbrace{W_V E}_{\textbf{Embedding}}\ \underbrace{\sigma(E^\top W_K^\top W_Q e^{(N+1)})}_{\textbf{Attention Module}}, \tag{10}$$

where the last term is the attention module. As is specified in Appendix A.1, with simple choices of $W_V, W_K, W_Q \in \theta$, one can show (10) subsumes (9); that is, Transformers can perform kernel regression in their forward pass. The key here is the connection between the kernel function and the attention matrix, which has been widely studied in previous literature, e.g. (Tsai et al., 2019; Wright & Gonzalez, 2021; Chen et al., 2024; Choromanski et al., 2021; Katharopoulos et al., 2020; Wang et al., 2020; Peng et al., 2021) and also discussed in the setting of ICL (Von Oswald et al., 2023; Cheng et al., 2024; Guo et al., 2024). In Appendix B, we provide a comprehensive discussion of their connections and how the feature map $\phi$ is related to Transformer parameter and architectural choices.

**Extension to $k > 1$.** To extend this result to CoT, we define $k$ loss functions $\{\mathcal{L}_i\}_{i \in [k]}$ associated with $k$ reasoning steps. Each loss function is convex w.r.t. the weights of the corresponding predictor. In the forward pass, similar with (9), Transformers implement (kernel) GD dynamics in function space to minimize these loss functions. One challenge of retaining the connection between (9) and (10) while further incorporating CoT is that, for compositional non-linear predictors $h$ in (8), updating weights in a prior-step predictors (e.g. $W_1$ in $h_1$) could introduce non-linear dynamics in the final prediction $h(x)$ from (8). We show that this issue can be circumvented if the learning is done in a step-by-step manner as in $\mathcal{A}_{\mathrm{CoT}}$, namely Transformers first learn a preceding reasoning step using $t$ layers, then proceed to learn the next step using $t$ layers. This results in a total of $kt$ layers for $k$ steps. Note that the construction here is not unique and similar conclusions could be drawn from other setups, such as recurrently making $k$ predictions (Li et al., 2023b) in $k$ forward passes, which would additionally require the Transformer to perform the so-called filtering process but could potentially reduce the depth requirement to a constant. Our construction differs from Von Oswald et al. (2023); Cheng et al. (2024) by extending their proof to the case where CoT is used.

### 3.3 Effects of Task Decomposition

We now proceed to answer when and how CoT improves the learnability of a task by studying the in-context learnability of a distribution family $\mathcal{P}$ w.r.t. different task decomposition schemes. Before presenting the main result, we analyze how well can the predictor $h$ generalizes to unseen queries. This is accomplished by studying the final error $\Delta(\mathcal{P}, \mathcal{A})$ made by the learning algorithm $\mathcal{A}_{\mathrm{CoT}}$ and its relation with individual errors made by individual algorithms at each step.

Consider a fixed number of steps $k$. According to Definition 2, applying a sequence of decomposition operators $\{T_i\}_{i \in [k-1]}$ on $\mathcal{P}$ produces $k$ new distribution families $\{\mathcal{P}_i\}_{i \in [k]}$, where $\mathcal{P}_i$ is the induced distribution family that generates the $i$-th reasoning step. As per equation (5), $\Delta(\mathcal{P}_i, \mathcal{A}_i)$ refers to the error of learning $\mathcal{P}_i$ using the individual algorithm $\mathcal{A}_i$. The following lemma upper bounds the final error $\Delta(\mathcal{P}, \mathcal{A})$ by the individual errors.

**Lemma 2** (Error Upper Bound). *Fix $k > 1$. Then for any distribution family $P$ and any decomposition operators $\{T_i\}_{i\in[k-1]}$ applied on $\mathcal{P}$, the predictor returned by $\mathcal{A}_{\mathrm{CoT}}$ has an error upper bound*

$$\Delta(\mathcal{P}, \mathcal{A}) \leq c_k \sum_{i=1}^{k} \Delta(\mathcal{P}_i, \mathcal{A}_i), \quad c_k = 2 \max_j \prod_{i=j+1}^{k} 2B^2 \operatorname{Lip}(\phi_i)^2 \tag{11}$$

*where $c_k$ is a constant depending on the hypothesis class $\mathcal{H}_i$ and the step number $k$.*

Note that this result does not rely on the specific individual algorithm we choose in $\mathcal{A}_{\mathrm{CoT}}$ as long as the predictor for each step has a Lipschitz constant and the loss function is convex. The upper bound suggests that, to minimize $\Delta(\mathcal{P}, \mathcal{A})$, it suffices for each individual algorithm to minimize the error made at its corresponding step. In fact, it actually suffices to minimize the largest error made at the hardest reasoning step $\operatorname{argmax}_i \{\Delta(\mathcal{P}_i, \mathcal{A}_i)\}$, as is revealed by the following result.

**Theorem 3** (Improved Learnability with CoT). *With CoT, any distribution family $\mathcal{P}$ is efficiently in-context learnable by the Transformer in Lemma 1, if there exists a finite sequence of decomposition operators $\{T_i\}_{i\in[k-1]}$ such that each induced $\mathcal{P}_i$ if efficiently PAC learnable by the individual algorithm $\mathcal{A}_i$. Particularly, the sample complexity is $N_{\mathcal{A}}(\epsilon, \delta) = \max_{i\in[k]} N_{\mathcal{A}_i}(\delta/k, \epsilon/c_k)$ where $N_{\mathcal{A}_i}$ is the sample complexity for learning $\mathcal{P}_i$ by $\mathcal{A}_i$.*

**Theoretical / Practical Implications.** This result indicates that, regardless of how complex the original reasoning task $\mathcal{P}$ is, in order for a Transformer to efficiently learn this task, it is sufficient to make each subtask $\mathcal{P}_i$ efficiently learnable. In other words, CoT can reduce the difficulty of learning a task to the difficulty of learning the hardest subtask in the chain; here, the hardest step refers to the least sample-efficient one. This result also leads to a simple practical lesson for designing CoT: *an effective way to form a CoT is by decomposing the hardest reasoning step into smaller steps that are easier to learn.* Such a result aligns with existing empirical practices of decomposing challenging tasks, e.g. (Zhou et al., 2022; Khot et al., 2022; Zhang et al., 2022), and will be validated in greater depth by new experiments in Section 5.1.

To understand this result, notice that in $\mathcal{A}_{\mathrm{CoT}}$ each subtask shares the same sample size, and thus one has to choose this size based on the hardest step to ensure each individual step can be successfully learned so that the overall task can be successful learned. Another way to view this is through the lens of error (Lemma 2): in the limit of large sample size $N$, the individual error at the step with the worst rate will dominate the overall error, regardless of the constant coefficient associated with each $\Delta(\mathcal{P}_{T,i}, h_i)$; therefore, the hardest step becomes the bottleneck.

**Compounding Error Issue.** Nevertheless, we note one caveat of Lemma 2 is that it is not asymptotic in $k$ which has been treated as a constant. Therefore, the result in this section does not hold if $k$ scales up, in which case errors could accumulate over reasoning steps and grow exponentially in the horizon. This implies a trade-off between the step number and hardness of subtasks: decomposing the hardest subtask improves the learnability, but also introduces the risk of compounding error, which renders scaling up $k$ a practical challenge despite the theoretical merit. Aligned with our theory, the compounding error issue has indeed been widely observed in practice, and approaches such as self-correction or refinement have been proposed to mitigate this issue, e.g. (Wang et al., 2023; Yao et al., 2024; Madaan et al., 2024). From our experiments in Section 5.1, we also find when the hardness of each reasoning step is approximately the same, increasing $k$ increases the overall task hardness and could hurt performance; however, for fixed overall task hardness, reducing the hardness of the hardest step can significantly improve the performance, which aligns with Theorem 3. We also note that the required GD steps $t$ for each step depends on parameters $\epsilon$ and $\delta$ and depth of the constructed Transformer in Lemma 1 could increase for harder tasks.

## 4 HARDNESS OF LEARNING WITHOUT CHAIN-OF-THOUGHT

In this section, we will present impossibility results demonstrating that there exist inherently hard tasks that are not learnable without CoT but learnable after being decomposed.

### 4.1 LOWER BOUND

We begin by presenting a general lower bound on the error $\Delta(\mathcal{P}, \mathcal{A})$. This bound applies to both the case where there is no CoT ($k = 1$), and the case of CoT with one intermediate step ($k = 2$) — each demonstration is in the form of $(x, z, y)$. The proof is based on the limited power of the hypothesis class $\mathcal{H}$ to approximate the target functions in $\mathcal{P}$ (see Appendix A.4).

**Theorem 4** (Error Lower Bound). *For any distribution family $\mathcal{P}$ and decomposition operator $T$, suppose $\mathcal{A}_{\mathrm{CoT}}$ returns a first-step predictor $h_1$ from a finite set $\mathcal{H}_1' \subseteq \mathcal{H}_1$. Then the overall error has a lower bound given by*[1]

$$\Delta(\mathcal{P}, \mathcal{A}) \geq \frac{1}{2} - B\sqrt{K|\mathcal{H}_1'|\,\mathrm{Var}(\mathcal{P})}, \tag{12}$$

*where $B$ and $K$ are constants depending on the definition of the hypothesis class as per (7). $\mathrm{Var}(\mathcal{P})$ is a certain variable depending on $\mathcal{P}$, we defer its definition to Appendix A.4.*

We discuss the implications of this theorem in two separate cases: when CoT is used and not used.

**w/ CoT.** In this case, the bound is generally loose or even uninformative, because the term $|\mathcal{H}_1'|$ is typically expected to be large. This makes sense since when CoT is used, the hypothesis class $\mathcal{H} = \mathcal{H}_2 \circ \mathcal{H}_1$ per equation (8) can potentially approximate a wide range of functions analogous to what a two-layer neural network can approximate. Despite this, the bound is still useful to identify cases where the CoT is suboptimal. For instance, consider a dummy CoT where the first step is the identity mapping (i.e., $x = z$), and assume the learner has successfully learned this step, which indicates that $h_1$ is also an identity mapping, and hence $|\mathcal{H}_1'| = 1$. In this case, the lower bound would increase and potentially become positive depending on $\mathrm{Var}(\mathcal{P})$ which will be discussed below, suggesting this dummy CoT is suboptimal.

**w/o CoT.** A more interesting scenario is when there is no CoT. In this case the lower bound from (12) reduces to $1/2 - B\sqrt{K\,\mathrm{Var}(\mathcal{P})}$ (Malach & Shalev-Shwartz, 2022), quantifying how hard it is to approximate $\mathcal{P}$ using the hypothesis class $\mathcal{H} = \{x \mapsto W\phi(x) : \|W\|_2 \leq B\}$ per definition in (7). The key quantity in the lower bound is $\mathrm{Var}(\mathcal{P})$, which indicates the intrinsic complexity of the task; the more complex $\mathcal{P}$ is, the smaller $\mathrm{Var}(\mathcal{P})$ is. In the next subsection, we will discuss learning a specific family of Boolean functions called parities; these functions underpin a concrete scenario whereby the task is unlearnable without CoT but becomes learnable when CoT is used.

## 4.2 ILLUSTRATIVE EXAMPLE: LEARNING PARITIES

Boolean functions are mappings from an input space $\mathcal{X} = \{\pm 1\}^n$ of $n$ binary bits to an output space $\mathcal{Y} = \{\pm 1\}$. In particular, parities are a family of functions that compute the exclusive-or (XOR) of bits at some predefined positions in the input, which are notoriously hard to learn (Kearns, 1998; Shalev-Shwartz et al., 2017; Daniely & Malach, 2020). The specific form of a parity function is determined by a subset $S \subseteq [n]$. For each $S$, the corresponding parity function is defined as $\chi_S(x) = \prod_{i \in S} x[i]$ where $x[i]$ is the $i$-th bit of the input. The distribution family is defined as

$$\mathcal{P}_{\mathrm{XOR}(n)} \triangleq \{(\chi_S, \mathcal{D}) : S \subseteq [n]\} \tag{13}$$

where $\mathcal{D}$ is a fixed input distribution uniform over $\{\pm 1\}^n$. Particularly, applying the lower bound to learning parities, we have the following result:

**Corollary 5** (Learning Parities). *For any hypothesis class defined in (7), there exists some sufficiently large $n \in \mathbb{N}$ such that $\mathcal{P}_{\mathrm{XOR}(n)}$ is not learnable by the corresponding algorithm $\mathcal{A}_{\mathrm{CoT}}$ when $k = 1$, but learnable after using the following decomposition to form the CoT ($k = 2$):*

$$\textbf{\textit{1st Step:}} \quad z[i] = \chi_{1,S}(x)[i] = \begin{cases} x[i] & \text{for } i \in S \\ 1 & \text{for } i \notin S \end{cases}, \quad \textbf{\textit{2nd Step:}} \quad y = \chi_{2,S}(z) = \prod_i z[i]. \tag{14}$$

*where the first step $\chi_{1,S}(x)$ learns to select relevant features from $x$ while masking irrelevant ones, and the second step $\chi_{2,S}(z)$ computes XOR of all bits in $z$.*

The proof is deferred to Appendix A.5. Intuitively, parities are hard to learn without CoT because the $2^n$ target functions in the family form an orthogonal basis for the space of all Boolean functions. Consequently, a linear function class with a fixed feature space dimension $K$ can not approximate all parities, as this is as hard as approximating all Boolean functions. Concretely, we have $\mathrm{Var}(\mathcal{P}_{\mathrm{XOR}(n)}) = 2^{-n}$ and thus the lower bound will become positive for sufficiently large $n$; this implies there always exists at least one target function in $\mathcal{P}_{\mathrm{XOR}(n)}$ and some $\epsilon_0 > 0$ such that $\epsilon \geq \epsilon_0$ holds true, and thus the task is not learnable per definition in Section 2 regardless of the sample size. CoT resolves this issue, since the first-step predictor with learnable weights can adapt to the relevant

---

[1]Note that in the case of no CoT, by default $|\mathcal{H}_1'| = 1$ and the bound still applies.

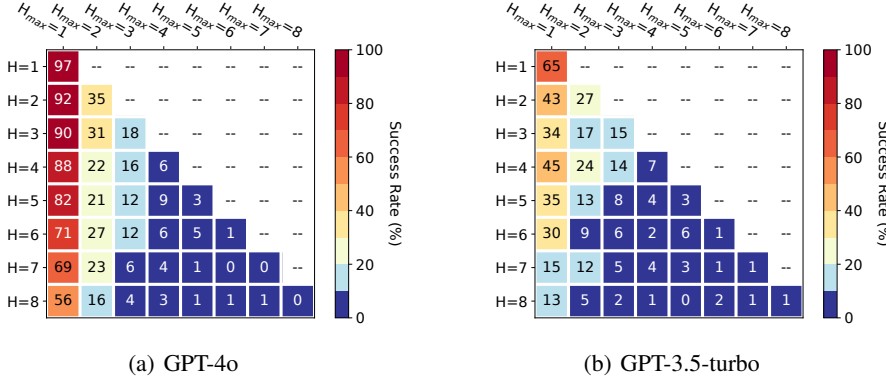

(a) GPT-4o  (b) GPT-3.5-turbo

Figure 2: Success rate of GPT-4o and GPT-3.5-turbo for learning compositional functions. $H$ denotes the number of elementary functions used to construct the target function; $H_{max}$ denotes the maximal number of elementary functions to construct a reasoning step.

bits in the input $x$ (determined by $S$), such that the second-step only needs to learn a fixed XOR function. In particular, with CoT, both steps or subtasks become learnable by a linear function class with a fixed feature space dimension. Therefore, we have illustrated such a setting where successful learning is impossible without CoT. The effectiveness of the designed CoT is verified in Section 5.2.

## 5 EXPERIMENTS

As empirical verification, we consider new arithmetic reasoning tasks and evaluate the performance of real-world LLMs, including GPT-4o and GPT-3.5-turbo. Sec 5.1 studies the connection between the hardest step and the overall reasoning performance. Sec 5.2 tests whether the specific forms of CoT we introduced can improve the performance. See detailed experimental setups in Appendix C. Codes are available at https://github.com/chr26195/CoT-ICL.

### 5.1 INCREASINGLY COMPLEX FUNCTIONS

Since benchmarks are lacking where one can precisely control the hardness of tasks and CoT steps, we first present a method to construct such tasks by incrementally building upon challenging subtasks.

**Constructing Highly Challenging Tasks.** Let us consider a class of elementary functions $\mathcal{F}_e$ where each function maps from the input space $\mathcal{X}$ to itself. In general, these elementary functions should be considered equally easy to learn. Then, we sample a sequence of these functions $f_1, f_2, \ldots, f_T \sim \mathcal{F}_e$; composing them gives us a target function $f = f_T \circ \cdots \circ f_2 \circ f_1 : \mathcal{X} \to \mathcal{X}$ whose complexity increases as $H$ increases. We consider an instantiation by defining the input space as the space of two integers $x \in \mathbb{Z}^2$. The elementary functions are defined as choosing one integer and using it to perform a basic arithmetic operation (drawn from $+$, $-$ or $\times$) with another number. Therefore, $\mathcal{F}_e$ consists of

$$
\begin{aligned}
&z[0] \leftarrow z[0] + z[1], \quad z[0] \leftarrow z[0] - z[1], \quad z[0] \leftarrow z[0] \times z[1], \\
&z[1] \leftarrow z[1] + z[0], \quad z[1] \leftarrow z[1] - z[0], \quad z[1] \leftarrow z[1] \times z[0].
\end{aligned}
\tag{15}
$$

While each elementary function in (15) is simple, the overall target function $f$ can become highly complex, possibly representing polynomial functions on $z[0]$ and $z[1]$ up to an arbitrary order and number of terms. Moreover, to quantify the hardest step, we do not reveal all intermediate steps of $f$ in the demonstrations provided to LLMs. Instead, we stipulate that there exists at least one step $i \in [k]$ where the function from $z_{i-1}$ to $z_i$ is constructed from $H_{max}$ elementary functions, whereas all other steps use fewer of them. For example, given $H = 3$ elementary functions $f_1 : z[0] \leftarrow z[0] + z[1]$, $f_2 : z[1] \leftarrow z[1] \times z[0]$ and $f_3 : z[1] \leftarrow z[1] - z[0]$, the hardest step can be expressed as $f_3 \circ f_2 \circ f_1 :$

$$
\begin{cases}
z_i[0] &= z_{i-1}[0] + z_{i-1}[1] \\
z_i[1] &= (z_{i-1}[0] + z_{i-1}[1]) (z_{i-1}[1] - 1)
\end{cases}
\tag{16}
$$

**Results.** We test the performance of real-world LLMs on the reasoning task described above with respect to different overall hardness $H$ and the hardest step $H_{max}$. We report their success rates

| $(n,k)$-Parities | (10,1) | (10,2) | (10,3) | (10,4) | (10,5) | (10,6) | (10,7) | (10,8) | (10,9) | (10,10) |
|---|---|---|---|---|---|---|---|---|---|---|
| GPT-4o w/o CoT | 87 | 69 | 63 | 54 | 51 | 48 | 50 | 52 | 47 | 51 |
| GPT-4o w CoT | **92** | **95** | **97** | **94** | **87** | **73** | **66** | 58 | **62** | 50 |
| GPT-3.5-turbo w/o CoT | 75 | 62 | 60 | 47 | 59 | 58 | 51 | 56 | 55 | 54 |
| GPT-3.5-turbo w CoT | 80 | 76 | 72 | 74 | 75 | 69 | 57 | **63** | 57 | **57** |

Table 1: Success rate (%) of GPT-4o and GPT-3.5-turbo of learning (n,k)-parities.

| 3-Term DNF | Width 3 | Width 4 | Width 5 | Width 6 | Width 7 | Width 8 | Width 9 | Width 10 |
|---|---|---|---|---|---|---|---|---|
| GPT-4o w/o CoT | 85 | 81 | 77 | 73 | 68 | 66 | 62 | 74 |
| GPT-4o w CoT | **96** | **87** | 86 | **88** | **81** | **80** | **80** | **84** |
| GPT-3.5-turbo w/o CoT | 74 | 68 | 67 | 62 | 55 | 58 | 64 | 53 |
| GPT-3.5-turbo w CoT | 90 | 78 | **87** | 81 | 65 | 73 | 70 | 73 |

Table 2: Success rate (%) of GPT-4o and GPT-3.5-turbo of learning 3-term DNF.

across 100 i.i.d. sampled target functions for each $H$ and $H_{max}$. And for each target function, the LLMs are provided with 10 demonstrations and asked to infer the computation process and apply it to derive the output for an unseen input. As shown in Fig. 2 (and more results in Appendix C.1), the success rate of LLMs quickly drops as $H_{max}$ increases. In particular, GPT-4o can successfully learn the target function in most cases when $H = 1$; however, it performs significantly worse as $H_{max}$ increases from 1 to 4, then fails as $H_{max}$ becomes even larger. These phenomena corroborate our result that reducing the complexity of the hardest step is critical to successfully handle the task.

## 5.2 CANONICAL BOOLEAN FUNCTIONS

We further evaluate LLMs on two families of Boolean functions: parities and disjunctive normal form (DNF), which are known hard to learn (Daniely & Vardi, 2021; Malach & Shalev-Shwartz, 2022).

**Task Descriptions.** An $(n, k)$-parity function computes the XOR ($\oplus$) of a subset of $k$ variables from a total of $n$ input binary bits ($n$ is 10 in our experiments). It outputs 1 if an odd number of the $k$ relevant variables are 1, and 0 otherwise. Meanwhile, a DNF function is a disjunction (logical OR) of conjunctions (logical ANDs) of literals; and in the experiments, we consider a family of 3-term DNFs $f(x) = \vee_{i=1}^{3} \wedge_{j=1}^{w} (x_{ij} \vee m_{ij})$ where $w$ is the width and $m \in \{\pm 1\}^{3w}$ is a latent variable whose value determines the target function (i.e. $m_{ij} = 1$ invalidates $x_{ij}$).

**Results.** For each $k$ in parities and $w$ in DNFs, we similarly i.i.d. sample 100 target functions.[2] For each function, we provide LLMs with 100 in-context examples, ask them to find patterns in these examples, and return the output for each query. Tables 1 and 2 report the success rates. Particularly n terms of parities, we find even GPT-4o generally performs no better than random guessing (with an expected accuracy of $50\%$) when $k > 3$. Then, we provide LLMs CoT examples with intermediate steps that are provably effective by applying our results derived in Section 4: for parities, the intermediate step is defined as $z[i] = x[i]$ if $i \in S$ otherwise 0; for DNFs, $z[i, j] = x[i, j]$ if $m[i, j] = 0$ otherwise 1. Results in Tables 1 and 2 clearly demonstrate the designed CoT significantly improves the performance, e.g. GPT-4o achieves an almost perfect success rate of 94 on $(10, 4)$-parity, while without CoT the success rate is 54, which is close to random guessing.

## 6 CONCLUSION AND DISCUSSION

In this paper, we quantify the benefits of task decompositions within the setting of learning tasks in-context by Transformers. We note that a limitation of our work thus far is that, despite quantifying the potential for a Transformer to efficiently learn a complex task by CoT, it nonetheless remains unclear on a case-by-case basis whether real-world LLMs will actually achieve success in practice. This is because of confounding issues related to model training procedures, including dataset properties, the actual optimization process, and fine-tuning. Hence an interesting future direction is to delve more deeply into these issues.

---

[2]For parities, we sample from a uniform distribution; for DNF, we sample from a non-uniform distribution to ensure the label (0/1) is balanced for $w \geq 3$.

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

CONTENTS

# A  PROOFS

## A.1  LEMMA 1: EXPRESSIVENESS OF TRANSFORMERS

Given demonstrations $\{(z_0^{(j)}, z_1^{(j)}, \cdots, z_{k-1}^{(j)}, z_k^{(j)})\}_{j=1}^N$, we could create $k$ training sets, each of which defines a loss function quantifying the error of a particular predictor $h_i$. These loss functions are

$$\left\{ \mathcal{L}_i = \frac{1}{2} \sum_{j=1}^N \left\| h_i(z_{i-1}^{(j)}) - z_i^{(j)} \right\|_2^2 : i \in [k] \right\}. \tag{17}$$

Note that while the overall predictor

$$h = h_k \circ \cdots \circ h_2 \circ h_1 = W_k \phi_k(\cdots (W_2 \phi_2(W_1 \phi_1(x)))) \tag{18}$$

is a non-linear function, loss functions in (17) are convex with respective to the weights of their corresponding linear predictors $\{W_i : i \in [k]\}$. Let us also denote $h_{\leq k'} = h_{k'} \circ \cdots \circ h_2 \circ h_1$ for $k' \leq k$. Using gradient descent to minimize $\mathcal{L}_i$ with fixed step size $\eta$ induces the following training dynamics in weight space

$$W_i \leftarrow W_i - \eta \nabla_{W_i} \mathcal{L}_i = W_i + \eta \left( Z_i - h_i(Z_{i-1}) \right) \phi_i(Z_{i-1})^\top \tag{19}$$

where $h_i(Z_{i-1}) = [h_i(z_{i-1}^{(j)})]_{j \in [N]} \in \mathbb{R}^{d(\mathcal{Z}_i) \times N}$, $\phi_i(Z_{i-1}) = [\phi_i(z_{i-1}^{(j)})]_{j \in [N]} \in \mathbb{R}^{K \times N}$. Note that difference between our setup and the conventional supervised learning setup is that, the latter is interested in the variation of output with different weights, while in this paper, we are also interested in the dynamics of intermediate steps. Particularly:

- For $i' < i$, the dynamics of intermediate steps induced by GD is

$$h_{\leq i'}(x^{(N+1)}) \leftarrow h_{\leq i'}(x^{(N+1)}), \tag{20}$$

namely the variation of upper layer weights does not affect lower layer representations (i.e. intermediate steps).

- For $i' = i$, the dynamics is

$$\begin{aligned}
h_{\leq i}(x^{(N+1)}) &= W_i \phi_i(h_{\leq i-1}(x^{(N+1)})) \\
&\leftarrow (W_i - \eta \nabla_{W_i} \mathcal{L}_i) \phi_i(h_{\leq i-1}(x^{(N+1)})) \\
&= h_{\leq i}(x^{(N+1)}) + \eta \left( Z_i - h_i(Z_{i-1}) \right) \phi_i(Z_{i-1})^\top \phi_i(h_{\leq i-1}(x^{(N+1)})).
\end{aligned} \tag{21}$$

Let $\kappa_i$ be the kernel function defined by the feature map $\phi_i$, we have

$$h_{\leq i}(x^{(N+1)}) \leftarrow h_{\leq i}(x^{(N+1)}) + \eta \left( Z_i - h_i(Z_{i-1}) \right) \kappa_i(Z_{i-1}, h_{\leq i-1}(x^{(N+1)})). \tag{22}$$

- For $i' > i$, the dynamics is

$$\begin{aligned}
h_{\leq i'}(x^{(N+1)}) &= h_i \circ \cdots \circ h_{i'+1} \circ h_{\leq i'}(x^{(N+1)}) \\
&\leftarrow h_i \circ \cdots \circ h_{i'+1} \left( h_{\leq i}(x^{(N+1)}) + \eta \left( Z_i - h_i(Z_{i-1}) \right) \kappa_i(Z_{i-1}, h_{\leq i-1}(x^{(N+1)})) \right)
\end{aligned} \tag{23}$$

which in general intractable since $h_i \circ \cdots \circ h_{i'+1}$ is non-linear. However, if upper layer weights in $h_i, \cdots, h_{i'+1}$ are 0, $h_{\leq i'}(x^{(N+1)})$ will become 0 as well and thus we can circumvent (23).

Recall also that based on the definition in Section 2, the self-attention layer can be written as

$$\text{(Self-Attention)} \qquad e^{(N+1)} \leftarrow e^{(N+1)} + W_V E \sigma \left( E^\top W_K^\top W_Q e^{(N+1)} \right). \tag{24}$$

where $e = (x, z_1, \cdots, z_{k-1}, z_k) \in \mathbb{R}^d$ at the input layer, $d = \sum_{i=0}^k d(\mathcal{Z}_i)$.

In the following construction, we show that in the forward pass of Transformer, (24) could express dynamics of all intermediate steps, including (20), (22) and (23), based on a certain order in which minimization of losses in (17) is performed. Particularly, in our construction, Transformer will sequentially minimize loss functions in (17). In other words, the lower layers of the Transformer learn prior reasoning steps, while upper layers of the Transformer learn later reasoning steps.

**Expanded Embedding Construction.** Let $d = \sum_{i=0}^{k} d(\mathcal{Z}_i)$ be the total dimension of the original demonstration $(z_0, z_1, \ldots, z_k)$ where $z_0 = x \in \mathcal{Z}_0$ and $z_k \in \mathcal{Z}_k$. Let $d' = 2d - d(\mathcal{Z}_0) - d(\mathcal{Z}_k)$ which will be the dimension of our expanded embedding. We now construct a projection matrix $P \in \mathbb{R}^{d' \times d}$ to map each demonstration into an expanded embeddings that keeps track of inputs and residuals in a single vector. Concretely, we partition $P$ to perform the following transformation:

$$
P = \begin{pmatrix}
I_{d(\mathcal{Z}_0)} & 0 & \cdots & 0 & 0 \\
0 & -I_{d(\mathcal{Z}_1)} & \cdots & 0 & 0 \\
0 & I_{d(\mathcal{Z}_1)} & \cdots & 0 & 0 \\
0 & 0 & -I_{d(\mathcal{Z}_2)} & \cdots & 0 \\
0 & 0 & I_{d(\mathcal{Z}_2)} & \cdots & 0 \\
\vdots & \vdots & \vdots & \ddots & \vdots \\
0 & 0 & 0 & \cdots & -I_{d(\mathcal{Z}_k)}
\end{pmatrix}, \tag{25}
$$

where each $I_{d(\mathcal{Z}_i)}$ denotes a $d(\mathcal{Z}_i) \times d(\mathcal{Z}_i)$ identity block (or negative identity block). Applying this $P$ to a vector $(z_0, z_1, \ldots, z_k) \in \mathbb{R}^d$ yields an expanded embedding

$$
e = P \left( z_0, z_1, \ldots, z_k \right)^\top \in \mathbb{R}^{d'}. \tag{26}
$$

At initialization, each $h_i(z_{i-1})$ is zero, so the rows with $-I_{d(\mathcal{Z}_i)}$ produce residuals $\left( h_i(z_{i-1}) - z_i \right)$ which initially reduce to $(0 - z_i) = -z_i$. Hence, for a demonstration $(x, z_1, z_2, \ldots, z_k)$ with $x = z_0$, we explicitly obtain

$$
e = \left( x, \; h_1(x) - z_1, \; z_1, \; \ldots, \; h_{k-1}(z_{k-2}) - z_{k-1}, \; z_{k-1}, \; h_k(z_{k-1}) - z_k \right) \in \mathbb{R}^{d'} \tag{27}
$$

Similarly, for the query $x^{(N+1)} \in \mathcal{Z}_0$, we have the expanded embedding

$$
e^{(N+1)} = \left( x^{(N+1)}, \; h_1(x^{(N+1)}), \; 0, \; \ldots, \; h_{\leq k-1}(x^{(N+1)}), \; 0, \; h_{\leq k}(x^{(N+1)}) \right) \in \mathbb{R}^{d'}, \tag{28}
$$

reflecting that for the query we store prediction $h_{\leq i}(x^{(N+1)})$ at each layer in its own coordinate, with zeros in place of $z_i$ or differences as needed to run gradient-based updates in the Transformer. Here we leverage the fact that $z_1, \cdots, z_{k-1}, z_k$ are all-0 vectors since they are unknown.

**Weight Matrices Construction.** For layers that minimize the $i$-th step's loss function $\mathcal{L}_i$, we construct Transformer weights in the corresponding self-attention layer as follows.

$$
W_V = \begin{pmatrix}
0_{d_l(i)} & 0 & 0 \\
0 & -\eta\, I_{d(\mathcal{Z}_i)} & 0 \\
0 & 0 & 0_{d_r(i)}
\end{pmatrix}, \tag{29}
$$

where

$$
d_l(i) = 2\sum_{j=0}^{i-1} d(\mathcal{Z}_j) - d(\mathcal{X}), \quad d_r(i) = 2\sum_{j=i}^{k} d(\mathcal{Z}_j) - d(\mathcal{Z}_k) - d(\mathcal{Z}_i),
$$

selecting the residual coordinates in the expanded embedding (i.e. $z_i - h_i(z_{i-1})$). This extracts those residuals, multiplies them by $-\eta$, and writes them into the sublayer prediction coordinate.

We let $W_K$ and $W_Q$ select the appropriate coordinates for $z_{i-1}$ (from demonstrations) and $h_{\leq i-1}(x^{(N+1)})$ (from the query):

$$
W_K = \begin{pmatrix}
0_{d'_l(i)} & 0 & 0 \\
0 & I_{d(\mathcal{Z}_{i-1})} & 0 \\
0 & 0 & 0_{d'_r(i)}
\end{pmatrix}, \tag{30}
$$

$$
W_Q = \begin{pmatrix}
0_{d''_l(i)} & 0 & 0 \\
0 & I_{d(\mathcal{Z}_{i-1})} & 0 \\
0 & 0 & 0_{d''_r(i)}
\end{pmatrix}, \tag{31}
$$

where

$$d'_l(i) = 2\sum_{j=0}^{i-1} d(\mathcal{Z}_j) - d(\mathcal{X}) - d(\mathcal{Z}_{i-1}), \quad d'_r(i) = 2\sum_{j=i}^{k} d(\mathcal{Z}_j) - d(\mathcal{Z}_k), \tag{32}$$

and similarly $d''_l(i)$, $d''_r(i)$ define the row/column slices for the query coordinate. Stacking $t$ self-attention layers (each with the above $W_V$, $W_K$, $W_Q$ for subtask $i$) carries out $t$ gradient steps on $\mathcal{L}_i$. Finally, iterating over $i = 1, \dots, k$ in ascending order completes minimization of all losses in equation 17, yielding the final predictor that includes all intermediate and final predictions.

Extension to kernel regression follows from Proposition 2 in Von Oswald et al. (2023) and Proposition 1 in Cheng et al. (2024). Refer also to Appendix B for a discussion of the connection between kernel and attention.

## A.2 Lemma 2: Upper Bound

Given a target function $f$ and an input distribution $\mathcal{D}(x)$, the algorithm $\mathcal{A}$ returns a predictor $h$ based on $N$ i.i.d. samples, whose expected error is defined as $\mathbb{E}_{x\sim\mathcal{D}}[l(h(x), f(x))]$. Let us first consider a single step of task decomposition.

**Lemma 6.** *For any distribution family $\mathcal{P}$ and decomposition operator $T$, the predictor returned by $\mathcal{A}_{\mathrm{CoT}}$ on demonstrations sampled from the corresponding distributions in $\mathcal{P}_1$ and $\mathcal{P}_2$ has an error upper bound*

$$\Delta(\mathcal{P}, \mathcal{A}) \le 2\max\{1, c_{B,\phi}\}(\Delta(\mathcal{P}_1, \mathcal{A}_1) + \Delta(\mathcal{P}_2, \mathcal{A}_2)) \tag{33}$$

*where $c_{B,\phi} = B^2 \operatorname{Lip}(\phi)^2$ is a constant determined by the hypothesis class $\mathcal{H}$ in (7).*

*Proof.* For a family of distributions $\mathcal{P}$, the error is defined as

$$\Delta(\mathcal{P}, \mathcal{A}) \triangleq \max_{(f,\mathcal{D})\in\mathcal{P}} \mathbb{E}_{x\sim\mathcal{D}}\left[l\left(h(x), f(x)\right)\right], \tag{34}$$

where we slightly abuse notation here as $h$ is also dependent on the distribution $(f, \mathcal{D})$ and the learning algorithm. For a certain decomposition operator $T$, the target function can be expressed as $f = f_2 \circ f_1$ and the predictor $h = h_2 \circ h_1$. We have

$$\Delta(\mathcal{P}, \mathcal{A}) = \max_{(f,\mathcal{D})\in\mathcal{P}} \mathbb{E}_{x\sim\mathcal{D}}\left[\frac{1}{2}\left((h_2 \circ h_1)(x) - (h_2 \circ f_1)(x) + (h_2 \circ f_1)(x) - f(x)\right)^2\right] \tag{35}$$

Suppose the feature map $\phi(x)$ for $h_2$ has Lipschitz constant

$$\operatorname{Lip}(\phi) = \sup_{x\neq x'} \frac{\|\phi(x) - \phi(x')\|_2}{\|x - x'\|_2}, \tag{36}$$

and by Jensen's inequality, we have

$$\Delta(\mathcal{P}, \mathcal{A}) \le \max_{(f,\mathcal{D})\in\mathcal{P}} \mathbb{E}_{x\sim\mathcal{D}}\left[\left((h_2 \circ h_1)(x) - (h_2 \circ f_1)(x)\right)^2 + \left((h_2 \circ f_1)(x) - f(x)\right)^2\right] \tag{37}$$

$$\le \max_{(f,\mathcal{D})\in\mathcal{P}} \mathbb{E}_{x\sim\mathcal{D}}\left[B^2 \operatorname{Lip}(\phi)^2\|h_1(x) - f_1(x)\|_2^2 + (h_2(z) - f_2(z))^2\right] \tag{38}$$

$$\le B^2 \operatorname{Lip}(\phi)^2 \max_{(f,\mathcal{D})\in\mathcal{P}} \mathbb{E}_{x\sim\mathcal{D}}\left[\|h_1(x) - f_1(x)\|_2^2\right] + \max_{(f,\mathcal{D})\in\mathcal{P}} \mathbb{E}_{x\sim\mathcal{D}}\left[(h_2(z) - f_2(z))^2\right] \tag{39}$$

where $B^2 \operatorname{Lip}(\phi)^2$ is a constant determined by the definition of hypothesis class. Given decomposition operator $T$, the distribution family can be decomposed into $\mathcal{P}_1$ and $\mathcal{P}_2$. It follows that

$$\Delta(\mathcal{P}, \mathcal{A}) \le 2B^2 \operatorname{Lip}(\phi)^2 \Delta(\mathcal{P}_1, \mathcal{A}_1) + 2\Delta(\mathcal{P}_2, \mathcal{A}_2) \tag{40}$$

Let $c_{B,\phi} = \max\{1, B^2 \operatorname{Lip}(\phi)^2\}$, we get the desired upper bound. $\square$

To extend this lemma to the case where there are more reasoning steps, i.e. $k > 2$, let

$$\text{\bf Individual Error:} \quad \Delta(\mathcal{P}_i, \mathcal{A}_i) = \Delta_i = \max_{(f, \mathcal{D}) \in \mathcal{P}_i} \mathbb{E}_{z_{i-1} \sim \mathcal{D}}[l(h_i(z_{i-1}) - z_i)]$$

$$\text{\bf Accumulated Error:} \quad \dot{\Delta}(\mathcal{P}, \{\mathcal{A}_j\}_{j \in [i]}) = \dot{\Delta}_i = \max_{(f, \mathcal{D}) \in \mathcal{P}} \mathbb{E}_{x \sim \mathcal{D}}[l((h_i \circ \cdots \circ h_1)(x) - z_i)].$$

$$(41)$$

$\Delta_i$ is the error of learning an individual reasoning step $i$, which is consistent with its definition in Section 2, $\dot{\Delta}_i$ is the accumulated error from the first step to step $i$. Note that we slightly abuse the notation here since $\Delta(\mathcal{P}, \mathcal{A})$ is actually equivalent to $\dot{\Delta}_k$. We also have $\dot{\Delta}_1 = \Delta_1$. Therefore

$$\dot{\Delta}_k = \max_{(f, \mathcal{D}) \in \mathcal{P}} \mathbb{E}_{x \sim \mathcal{D}} \left[ \frac{1}{2} \|(h_i \circ \cdots \circ h_1)(x) - z_i\|_2^2 \right] \tag{42}$$

$$= \max_{(f, \mathcal{D}) \in \mathcal{P}} \mathbb{E}_{x \sim \mathcal{D}} \left[ \frac{1}{2} \|(h_i \circ \cdots \circ h_1)(x) - h_i(z_{i-1}) + h_i(z_{i-1}) - z_i\|_2^2 \right] \tag{43}$$

$$\leq 2B^2 \operatorname{Lip}(\phi_k)^2 \cdot \dot{\Delta}_{k-1} + 2\Delta_k \tag{44}$$

$$\leq \cdots$$

$$\leq 2 \sum_{j=1}^{k} \left( \prod_{i=j+1}^{k} 2B^2 \operatorname{Lip}(\phi_i)^2 \right) \Delta_j \tag{45}$$

We use the lemma from (43) to (44). Therefore, we have the following upper bound

$$\Delta(\mathcal{P}, \mathcal{A}) = \dot{\Delta}_k \leq c_k \sum_{i=1}^{k} \Delta(\mathcal{P}_i, \mathcal{A}_i) \tag{46}$$

where $c_k = 2 \max_j \prod_{i=j+1}^{k} 2B^2 \operatorname{Lip}(\phi_i)^2$ is a constant depending on the hypothesis class $\mathcal{H}_i$ and $k$.

Note that in the derivation, we could remove $\max_{(f, \mathcal{D}) \in \mathcal{P}}$ and directly analyze the expected error $\mathbb{E}_{x \sim \mathcal{D}}[l(h(x), f(x))]$, which will give us a similar result: the final error is upper bounded by the sum of individual errors up to a constant.

### A.3 THEOREM 3: CoT IMPROVES LEARNABILITY

Recall the definition of in-context learnability: we say a parametric model $\mathcal{M}$ can learn task $\mathcal{P}$ if there exists a learning algorithm $\mathcal{A}$ and a function $N : (0, 1)^2 \to \mathbb{N}$, such that for any confidence and accuracy parameters $\delta, \epsilon \in (0, 1)$:

1. Under a certain parameter choice $\theta$, we have $\mathcal{M}_\theta(\cdot, x) = \mathcal{A}(\cdot)(x)$ for any query $x$.

2. Given $N_\mathcal{A}(\delta, \epsilon)$ i.i.d. examples, the learning algorithm $\mathcal{A}$ returns a predictor $h$ such that with probability of at least $1 - \delta$, $\Delta(\mathcal{P}, \mathcal{A}) \leq \epsilon$.

Moreover, we say $\mathcal{P}$ can be efficiently in-context learned by $\mathcal{M}$ if both the running time of $\mathcal{A}$ and the sample size $N_\mathcal{A}(\delta, \epsilon)$ are polynomial in $\delta^{-1}$ and $\epsilon^{-1}$.

To see how it can be reduced to the learnability of subtasks after using CoT, recall the definition of PAC learnability:

**Definition 3.** *We say a subtask $\mathcal{P}_i$ can be efficiently learned by an algorithm $\mathcal{A}_i$ if its sample complexity and time complexity scale as $\operatorname{poly}(\delta_i^{-1}, \epsilon_i^{-1})$, where $\delta_i$ and $\epsilon_i$ are confidence and accuracy parameter for the subtask.*

Since by Lemma 1, we know there exists such a parameter choice such that $\operatorname{TF}_\theta(\cdot, x) = \mathcal{A}_{\text{CoT}}(\cdot)(x)$, our goal is then to show: if every subtask $\mathcal{P}_i$ is efficiently learnable by its corresponding individual algorithm $\mathcal{A}_i$, the overall task is also efficiently learnable by $\mathcal{A}_{\text{CoT}}$.

By Lemma 2, we know that for fixed $k$, the accuracy parameter $\epsilon$ of the overall task is upper bounded by the sum of the accuracy parameters of subtasks $\{\epsilon_i\}_{i=1}^{k}$ up to a constant coefficient. Additionally,

notice that, given $n$ (not necessarily independent) events $\mathcal{A}_1, \mathcal{A}_2, \ldots, \mathcal{A}_n$, each occurring with probability $P(\mathcal{A}_i) = 1 - \delta_i$, we have

$$P\left(\bigcap_{i=1}^{n} \mathcal{A}_i\right) \geq \max\left(0, 1 - \sum_{i=1}^{n} \delta_i\right). \tag{47}$$

This means the confidence parameter of the overall task $\delta$ is also upper bounded by the sum of confidence parameters of all subtasks $\{\delta_i\}_{i=1}^{k}$. Namely

$$\epsilon \leq c_k \sum_{i=1}^{k} \epsilon_i, \quad \delta \leq \sum_{i=1}^{k} \delta_i \tag{48}$$

In terms of the **time complexity** of $\mathcal{A}_{\text{CoT}}$, since each individual algorithm $\mathcal{A}_i$ runs in time polynomial in $\delta_i^{-1}$ and $\epsilon_i^{-1}$, we have $\text{Time}(\mathcal{A}_i) = \text{poly}(\delta_i^{-1}, \epsilon_i^{-1})$. Moreover, one can choose $\delta_1 = \delta_2 = \cdots = \delta_k = \delta', \epsilon_1 = \epsilon_2 = \cdots = \epsilon_k = \epsilon'$, correspondingly we have $\delta' \geq \delta/k$ and $\epsilon' \geq \epsilon/c_k$. Since $\mathcal{A}_{\text{CoT}}$ is a simple combination of all individual algorithms $\{\mathcal{A}_i\}_{i=1}^{k}$, we have

$$\text{Time}(\mathcal{A}_{\text{CoT}}) = \sum_{i=1}^{k} \text{Time}(\mathcal{A}_i) \leq \text{poly}(\delta^{-1}, \epsilon^{-1}), \tag{49}$$

meaning the learning algorithm is also computationally efficient.

In terms of the **sample complexity** of $\mathcal{A}_{\text{CoT}}$, we also let $\delta_1 = \delta_2 = \cdots = \delta_k = \delta' \geq \delta/k$ and $\epsilon_1 = \epsilon_2 = \cdots = \epsilon_k = \epsilon' \geq \epsilon/c_k$. Note that in the case of CoT, all individual algorithms use the same number of samples as the algorithm $\mathcal{A}_{\text{CoT}}$. To show the algorithm is sample-efficient, we can simply choose $N_{\mathcal{A}}(\delta, \epsilon)$ to be the largest among $\{N_{\mathcal{A}_i}(\delta', \epsilon')\}_{i=1}^{k}$, i.e.

$$N_{\mathcal{A}}(\delta, \epsilon) = \max_{i \in [k]} N_{\mathcal{A}_i}(\delta', \epsilon') \tag{50}$$

which ensures that $\mathcal{A}_{\text{CoT}}$ can achieve high accuracy with high probability. Since any $N_{\mathcal{A}_i}(\delta_i, \epsilon_i)$ is polynomial in $\delta'^{-1}$ and $\epsilon'^{-1}$, it holds that $N_{\mathcal{A}}(\delta, \epsilon)$ is also polynomial in $\delta^{-1}$ and $\epsilon^{-1}$. However, if $N_{\mathcal{A}}(\delta, \epsilon) < \max_{i \in [k]} N_{\mathcal{A}_i}(\delta', \epsilon')$, we have no guarantee that each step can successfully learn their corresponding reasoning step. *From here, we also proved that the sample efficiency of the overall step is that of the hardest step in the CoT.*

Since $\mathcal{A}_{\text{CoT}}$ can efficiently learn $\mathcal{P}$, and Transformer with linear depth and constant embedding size can express $\mathcal{A}_{\text{CoT}}$, we complete the proof.

### A.4 THEOREM 4: LOWER BOUND

The in-context learning error has approximation error lower bound, that is the minimum error achievable by a predictor in the hypothesis class $\mathcal{H} = \{h_2 \circ h_1 : h_1 \in \mathcal{H}'_1, h_2 \in \mathcal{H}_2\}$

$$\max_{(f, \mathcal{D}) \in \mathcal{P}} \mathbb{E}_{x \sim \mathcal{D}}\left[l\left(h(x), f(x)\right)\right] \geq \max_{(f, \mathcal{D}) \in \mathcal{P}} \min_{(h_1, h_2) \in \mathcal{H}'_1 \times \mathcal{H}_2} \mathbb{E}_{x \sim \mathcal{D}}\left[l\left(h(x), f(x)\right)\right]. \tag{51}$$

Thus it suffices to lower bound the approximation error.

To do so, notice that the learned first-step predictor $h_1$ is from finite function class $\mathcal{H}'_1$, which is a subset of the initial hypothesis class $\mathcal{H}_1$. We show the approximation power of $h(x)$ with finite-sized $\mathcal{H}'_1$ is lower bounded by a linear class whose size depends on $\mathcal{H}'_1$. In particular, suppose the hypothesis class is

$$\mathcal{H}'_1 = \{h_{1,1}, h_{1,2}, \cdots, h_{1,|\mathcal{H}'_1|}\}, \tag{52}$$

and based on the index of $h_1$ in $\mathcal{H}'_1$, the predictor $h(x)$ can be re-written as

$$h(x, j) \triangleq W_2(\phi \circ h_{1,j})(x) = \sum_{i=1}^{K} W_{2,i} \phi_{i,j}(x) \tag{53}$$

$$= \sum_{i=1}^{K} \sum_{i'=1}^{|\mathcal{H}'_1|} U_{i,i'} \phi_{i,i'}(x) \tag{54}$$

where $\phi_{i,j} = \phi_i \circ h_{1,j}$ and $U_{i,i'} = W_{2,i}$ if $i' = j$ otherwise 0. As (54) is an inner product of weight vector $U \in \mathbb{R}^{K|\mathcal{H}_1'|}$ and feature vector $\phi(x) \in \{\pm 1\}^{K|\mathcal{H}_1'|}$ in an expanded space, $h(x,j)$ reduces to a linear function.

Since the squared loss $l(h(x,j), f(x))$ is convex w.r.t. $U$ for arbitrary $x$ and $j$, its expectation $L_{f,\mathcal{D}}(h) \triangleq \mathbb{E}_{x \sim \mathcal{D}} [l(h(x,j), f(x))]$ is also convex w.r.t. $U$. Moreover, $h(x,j) = 0$ when $W_2$ or $U$ goes to 0. Therefore, given any $(f, \mathcal{D}) \in \mathcal{P}$ and any predictor $h$ with fixed $W_2$ and $j$ (and thus fixed $U$), by first-order condition, we have

$$L_{f,\mathcal{D}}(h) \geq L_{f,\mathcal{D}}(0) + \langle U - 0, \nabla_U L_{f,\mathcal{D}}(h)|_{U=0} \rangle \tag{55}$$

$$\geq \frac{1}{2} - \|U\|_2 \|\nabla_U L_{f,\mathcal{D}}(h)|_{U=0}\|_2 \tag{56}$$

where the last equation uses the fact $L_{f,\mathcal{D}}(0) = \mathbb{E}_{x \sim \mathcal{D}} [l(0, f(x))] = \frac{1}{2}$ and inequality $\langle v, u \rangle \geq -\|u\|_2 \|v\|_2$.

Notice $U$ has the same norm as $W_2$ and thus $\|U\|_2 \leq B$. Moreover, we have

$$\nabla(f, \mathcal{D}) \triangleq \|\nabla_U L_{f,\mathcal{D}}(h)|_{U=0}\|_2^2 = \|\nabla_U \mathbb{E}_{x \sim \mathcal{D}} [l(h(x,j), f(x))]|_{U=0}\|_2^2 \tag{57}$$

$$= \|\mathbb{E}_{x \sim \mathcal{D}} [\nabla_U l(h(x,j), f(x))|_{U=0}]\|_2^2 \tag{58}$$

$$= \left\|\mathbb{E}_{x \sim \mathcal{D}} \left[\nabla_U \frac{1}{2}(\langle U, \phi(x) \rangle - f(x))^2|_{U=0}\right]\right\|_2^2 \tag{59}$$

$$= \|\mathbb{E}_{x \sim \mathcal{D}} [\phi(x) f(x)]\|_2^2 \tag{60}$$

$$= \sum_{i=1}^{K} \sum_{j=1}^{|\mathcal{H}_1'|} \mathbb{E}_{x \sim \mathcal{D}} [\phi_{i,j}(x) f(x)]^2 \tag{61}$$

Subjecting it to (56) gives us that, for any $(f, \mathcal{D}) \in \mathcal{P}$ and any predictor $h(x)$ obtained from in-context learning, i.e.

$$\min_{(h_1, h_2) \in \mathcal{H}_1' \times \mathcal{H}_2} L_{f,\mathcal{D}}(h) \geq \frac{1}{2} - B \cdot \nabla(f, \mathcal{D})^{\frac{1}{2}} \tag{62}$$

Following from (51), the in-context learning error has lower bound

$$\Delta(\mathcal{P}, \mathcal{A}) \geq \max_{(f,\mathcal{D}) \in \mathcal{P}} \min_{(h_1, h_2) \in \mathcal{H}_1' \times \mathcal{H}_2} L_{f,\mathcal{D}}(h) \tag{63}$$

$$\geq \max_{(f,\mathcal{D}) \in \mathcal{P}} \left[\frac{1}{2} - B \cdot \nabla(f, \mathcal{D})^{\frac{1}{2}}\right] \tag{64}$$

$$\geq \mathbb{E}_{(f,\mathcal{D}) \in \mathcal{P}} \left[\frac{1}{2} - B \cdot \nabla(f, \mathcal{D})^{\frac{1}{2}}\right] \tag{65}$$

$$= \frac{1}{2} - B \cdot \mathbb{E}_{(f,\mathcal{D}) \in \mathcal{P}} \left[\nabla(f, \mathcal{D})^{\frac{1}{2}}\right] \tag{66}$$

By noting $\mathbb{E}[X^{\frac{1}{2}}] \leq \mathbb{E}[X]^{\frac{1}{2}}$, we have

$$\Delta(\mathcal{P}, \mathcal{A}) \geq \frac{1}{2} - B \cdot \mathbb{E}_{(f,\mathcal{D}) \in \mathcal{P}} [\nabla(f, \mathcal{D})]^{\frac{1}{2}} \tag{67}$$

$$= \frac{1}{2} - B \cdot \mathbb{E}_{(f,\mathcal{D}) \in \mathcal{P}} \left[\sum_{i=1}^{K} \sum_{j=1}^{|\mathcal{H}_1'|} \mathbb{E}_{x \sim \mathcal{D}} [\phi_{i,j}(x) f(x)]^2\right]^{\frac{1}{2}} \tag{68}$$

Let

$$\text{Var}(\mathcal{P}) \triangleq \sup_{\phi} \mathbb{E}_{(f,\mathcal{D}) \in \mathcal{P}} \left[\mathbb{E}_{x \sim \mathcal{D}} [\phi(x) f(x)]^2\right], \tag{69}$$

which is determined by target functions and distributions in $\mathcal{P}$, and could be understood as the intrinsic complexity of the distribution family (i.e. the more complex $\mathcal{P}$ is, the smaller $\text{Var}(\mathcal{P})$ is). It

follows that,

$$\Delta(\mathcal{P}, \mathcal{A}) \geq \frac{1}{2} - B\sqrt{\sum_{i=1}^{K}\sum_{j=1}^{|\mathcal{H}_1'|} \mathbb{E}_{(f,\mathcal{D})\in\mathcal{P}}\left[\mathbb{E}_{x\sim\mathcal{D}}[\phi_{i,j}(x)f(x)]^2\right]} \tag{70}$$

$$\geq \frac{1}{2} - B\sqrt{K|\mathcal{H}_1'|\operatorname{Var}(\mathcal{P})}, \tag{71}$$

completing the proof.

### A.5 COROLLARY 5: LEARNING PARITIES

**w/o CoT.** The distribution family of parities with input size $n$ is defined as

$$\mathcal{P}_{\text{XOR}(n)} \triangleq \{(\chi_S, \mathcal{D}) : S \subseteq [n]\} \quad \text{where} \quad \chi_S(x) = \prod_{i\in S} x[i]. \tag{72}$$

Based on (69), we can derive an upper bound on $\operatorname{Var}(\mathcal{P})$ for parities, i.e.

$$\operatorname{Var}(\mathcal{P}_{\text{XOR}(n)}) = \sup_{\phi} \mathbb{E}_{(\chi,\mathcal{D})\in\mathcal{P}_{\text{XOR}(n)}}\left[\mathbb{E}_{x\sim\mathcal{D}}[\phi(x)\chi(x)]^2\right] \tag{73}$$

$$= \sup_{\phi} \frac{1}{|\mathcal{P}_{\text{XOR}(n)}|} \sum_{S\subseteq[n]} \mathbb{E}_{x\sim\mathcal{D}}[\phi(x)\chi_S(x)]^2 \tag{74}$$

where $|\mathcal{P}_{\text{XOR}(n)}| = 2^n$ and $2^{[n]}$ is the power set of $[n]$. Note that $\{\chi_S : S \subseteq [n]\}$ forms a Fourier basis of Boolean functions, meaning that for any pair of different subsets $S_1, S_2 \subseteq [n]$, we have

$$\mathbb{E}_{x\sim\mathcal{D}}[\chi_{S_1}(x)\chi_{S_2}(x)] = 0. \tag{75}$$

Therefore $\mathbb{E}_{x\sim\mathcal{D}}[\phi(x)\chi_S(x)]$ is exactly the Fourier coefficient of Boolean function $\phi(x)$ corresponding to $S$, which we denote as $\widehat{\phi}(S)$. Therefore

$$\operatorname{Var}(\mathcal{P}_{\text{XOR}(n)}) = \frac{1}{2^n} \sum_{S\subseteq[n]} \widehat{\phi}(S)^2 = \frac{1}{2^n}. \tag{76}$$

Since, when there is no CoT, the error lower bound is

$$\Delta(\operatorname{Var}(\mathcal{P}_{\text{XOR}(n)}), h) \geq \frac{1}{2} - B\sqrt{K\operatorname{Var}(\mathcal{P}_{\text{XOR}(n)})} = \frac{1}{2} - B\sqrt{\frac{K}{2^n}}, \tag{77}$$

we can choose $n > 2 + \log_2 B^2 K$ such that this lower bound is always positive (regardless of the sample size).

**w/ CoT.** Consider the following CoT

**1st Step:** $z[i] = \chi_{1,S}(x)[i] = \begin{cases} x[i] & \text{for } i \in S \\ 1 & \text{for } i \notin S \end{cases}$, **2nd Step:** $y = \chi_{2,S}(z) = \prod_{i} z[i],$ (78)

we show each step can be approximated by the hypothesis class defined in (7)

$$\mathcal{H}_i = \{z_{i-1} \mapsto W_i\phi_i(z_{i-1}) : \|W_i\|_2 \leq B\} \tag{79}$$

where $\phi_i : \mathcal{Z}_{i-1} \to \mathbb{R}^K$ is a non-linear feature map to a $K$-dimensional space, and $W_i \in \mathbb{R}^{d(\mathcal{Z}_i)\times K}$ are learnable weights. We do this by construction. For the first-step predictor $\chi_{1,S}(x)$, we let $\phi_1(x) = (x, 1)$ be the connection of $x$ and 1; for the second-step predictor $\chi_{2,S}(x)$, we let $\phi_2(z) = \prod_i z[i]$. With these features, both steps can be learned by a simple linear model $Wx$, which also means the expected error could be arbitrarily small with sufficiently large sample size. The sample complexity follows the standard analysis for linear regression.

In contrast, when there is no CoT, $\Delta(\mathcal{P}, \mathcal{A})$ has a positive lower bound, meaning there always exists at least one $(f, \mathcal{D}) \in \mathcal{P}$ such that the expected error is greater than the lower bound.

## B   BACKGROUND: CONNECTION BETWEEN ATTENTION AND KERNEL

As the background knowledge for understanding the construction of Transformers in Lemma 1, here we provide a non-exhaustive summary of the connections between the attention matrix in Transformers and the kernel method from previous works (Von Oswald et al., 2023; Cheng et al., 2024; Guo et al., 2024; Tsai et al., 2019; Wright & Gonzalez, 2021; Chen et al., 2024). We rewrite the kernel gradient descent dynamics and the Transformer layer here for reference

$$\textbf{(GD Dynamics)} \quad h(x^{(N+1)}) \leftarrow \underbrace{h(x^{(N+1)})}_{\textbf{Predictions}} + \eta \underbrace{(Y - \hat{Y})}_{\textbf{Residuals}} \underbrace{\phi(X)^\top \phi(x^{(N+1)})}_{\textbf{Kernel Function}}, \tag{80}$$

$$\textbf{(Transformer Layer)} \quad e^{(N+1)} \leftarrow \underbrace{e^{(N+1)}}_{\textbf{Skip Connection}} + \underbrace{W_V E}_{\textbf{Embedding}} \underbrace{\sigma(E^\top W_K^\top W_Q e^{(N+1)})}_{\textbf{Attention Module}}, \tag{81}$$

Here, the kernel is induced by $\kappa(x, y) = \phi(x)^\top \phi(y)$. The definition of $\sigma$ could be flexibly chosen depending on the practical implementation of Transformers.

**In-Context Learning.**  Von Oswald et al. (2023) (Proposition 1) demonstrates in the most simple case where both the non-linearity in Transformer $\sigma$ and the feature map $\phi$ are identity mappings, the weight constructions $W_V = \begin{pmatrix} 0_{d(\mathcal{X})} & 0 \\ 0 & -\eta I_{d(\mathcal{Y})} \end{pmatrix}$ and $W_K^\top W_Q = \begin{pmatrix} I_{d(\mathcal{X})} & 0 \\ 0 & 0_{d(\mathcal{Y})} \end{pmatrix}$ yields

$$E^\top W_K^\top W_Q e^{(N+1)} = X^\top x_{N+1}, \quad W_V E = -\eta(0_{d(\mathcal{X})}, Y - \hat{Y}). \tag{82}$$

In this case, $\kappa(x, y) = x^\top y$ is the inner product kernel. Their Proposition 2 further discusses the case which accommodates the role of the MLP module in the Transformer architecture. Specifically the MLP module transforms the token $e$ as $\text{MLP}_\theta(e)$, and thus with the same $W_V, W_K, W_Q$ and identty mapping $\sigma$, the kernel is

$$\kappa(x, y) = \text{MLP}_\theta(x)^\top \text{MLP}_\theta(y). \tag{83}$$

Guo et al. (2024) also leverages the representation power of MLP and considers the case where $\phi: \mathbb{R}^d \to \mathbb{R}^K$ is a fixed representation function, which can in principle be chosen arbitrarily as long as the features can be represented by a MLP. Theorem 1 in this paper provides a construction where $\sigma$ is chosen to be the normalized ReLU and proves that Transformers can perform in-context ridge regression. Following this work, more recently Kim & Suzuki (2024) similarly concluded that MLP layer can extend the class of learnable functions of ICL to the Barron space.

Cheng et al. (2024) considers the case where $\sigma$ is non-linear, and their Proposition 1 proves if the non-linearity $\sigma$ matches the kernel $\kappa$, the Transformers can perform functional (kernel) gradient descent regression w.r.t. the reproducing kernel Hilbert space metric of this kernel; the proof is exactly based on the relation between (80) and (81). For example, they demonstrate that the $\text{Softmax}$ attention corresponds to the exponential kernel

$$\kappa(x, y) = \exp\left(-\frac{1}{\sigma^2} x^\top y\right). \tag{84}$$

One caveat here is that the attention matrix in this case is asymmetric; to address this, one could treat the normalization term as pre-conditioner of the optimization algorithm, and the construction of Transformers should be modified accordingly. We refer the readers to more discussions in their paper.

**General Discussions.**  Even before ICL, the connection between kernel and attention is a topic that has been widely discussed (e.g. (Tsai et al., 2019; Wright & Gonzalez, 2021; Chen et al., 2024) etc.) and references therein). For instance, Wright & Gonzalez (2021) proves that the standard attention matrix is a reproducing kernel for a reproducing kernel Banach space and gives explicit formulation of the feature maps (Proposition 1). Their Theorem 2 further demonstrates that Transformers can learn any binary non-Mercer reproducing kernel Banach space pair.

In practice, variants of Transformers with kernelized attentions (e.g. (Choromanski et al., 2021; Katharopoulos et al., 2020; Wang et al., 2020; Peng et al., 2021) etc.) such as those relying on random

Fourier features are very popular and effective. These implementations are often considered for efficiency considerations, since once the attention is kernelized, one can switch the order of matrix multiplication to accelerate the computation and reduce the time complexity from quadratic to linear w.r.t. the input length.

## C EXPERIMENTAL DETAILS AND ADDITIONAL RESULTS

### C.1 INCREASINGLY COMPLEX FUNCTIONS

**Sampling.** For this task, we sample a sequence of functions from a set of elementary functions with the following probabilities:

$$P\left(z[0] \leftarrow z[0] + z[1]\right) = \frac{1}{8}, \quad P\left(z[0] \leftarrow z[0] - z[1]\right) = \frac{1}{8}, \quad P\left(z[0] \leftarrow z[0] \times z[1]\right) = \frac{1}{4},$$

$$P\left(z[1] \leftarrow z[1] + z[0]\right) = \frac{1}{8}, \quad P\left(z[1] \leftarrow z[1] - z[0]\right) = \frac{1}{8}, \quad P\left(z[1] \leftarrow z[1] \times z[0]\right) = \frac{1}{4}.$$

Here, the multiplication operation is more likely to be sampled than addition or subtraction. For the input values, we uniformly select two unique integers from the set $\{2, 3, \ldots, 10\}$. This range is chosen to avoid excessively large numbers, which are difficult to handle, and trivial cases, such as when $x[0] = x[1]$, which could result in zero values that are easily predictable by LLMs. For this task, we explicitly ensure that each sample is unique to prevent repeated samples in training and testing. This is done to avoid scenarios where LLMs could simply memorize the results from the demonstrations and use them to answer the query.

**CoT Prompting.** Given a certain $H_{max}$, namely the maximal number of elementary functions to construct a reasoning step, we implement it by randomly masking $H - 1$ consecutive intermediate steps. For example, when $T = 6$ and $H = 3$, an example prompt is given as follows:

```
Given two numbers, sequentially apply predefined arithmetic
↪   operations (addition, subtraction, multiplication) to
↪   transform them. Each step involves a specific predefined
↪   operation on one of the numbers. If any operations or
↪   intermediate results are missing, deduce these to complete the
↪   transformation and arrive at the final output.

Input: 7, 5
Step1: 7, -2
Step2: 5, -2
Step3: missing
Step4: missing
Step5: 5, 75
Output: -70, 75

Input: 2, 3
Step1: 2, 1
Step2: 3, 1
Step3: missing
Step4: missing
Step5: 3, 36
Output: -33, 36

...

Input: 5, 8
What is the output? Your answer should end in the format 'Step1:
↪   ?, Step2: ?, ..., Output:?'.
```

Note that the missing steps are consistent for one trial. We test 100 times to compute the success rate.

**Additional Results.** In addition to reporting the success rate of LLMs for predicting the final output as presented in Section 5, we also evaluate their success rate for predicting intermediate steps. This provides a more comprehensive assessment of the LLMs' performance, as even they might fail to predict the final step but could still succeed in predicting the intermediate steps.

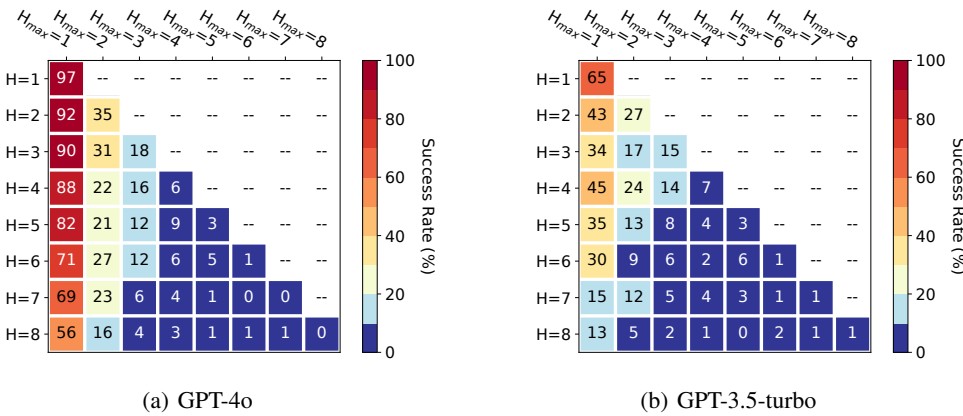

(a) GPT-4o

(b) GPT-3.5-turbo

Figure 3: Success rate for predicting the last step (i.e. namely the output).

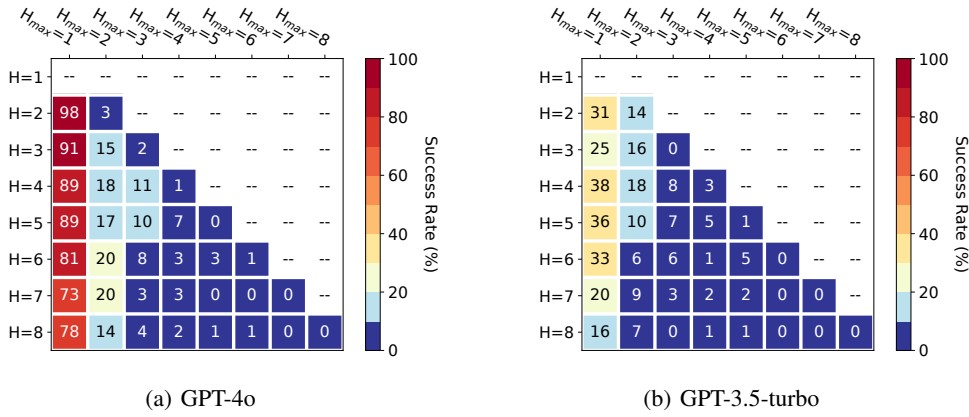

(a) GPT-4o

(b) GPT-3.5-turbo

Figure 4: Success rate for predicting the second to last step.

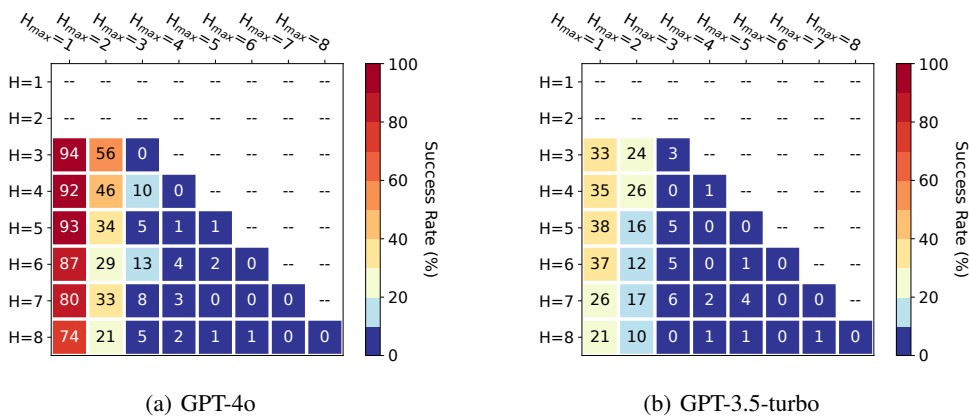

(a) GPT-4o

(b) GPT-3.5-turbo

Figure 5: Success rate for predicting the third to last step.

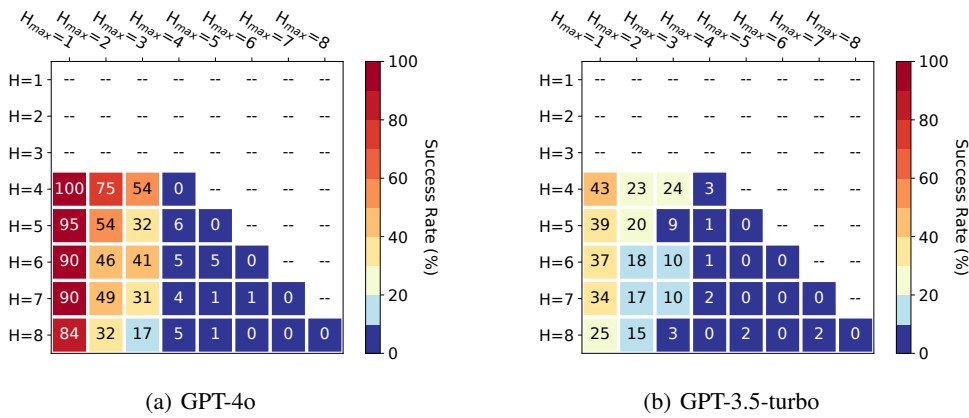

Figure 6: Success rate for predicting the fourth to last step.

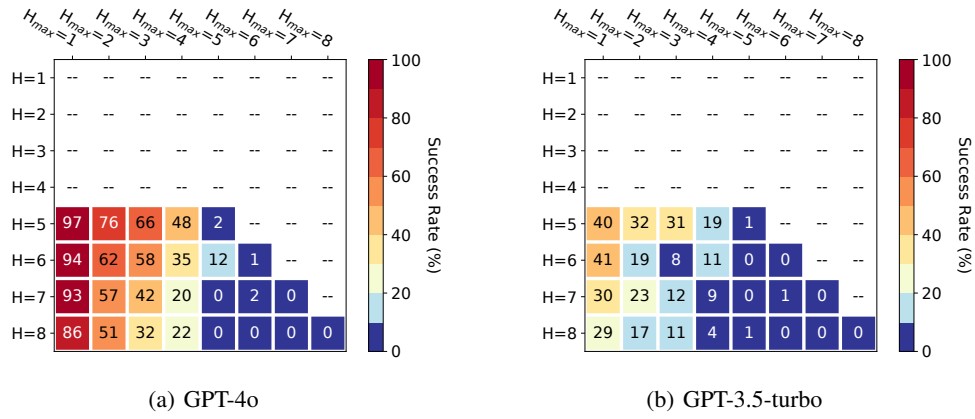

Figure 7: Success rate for predicting the fifth to last step.

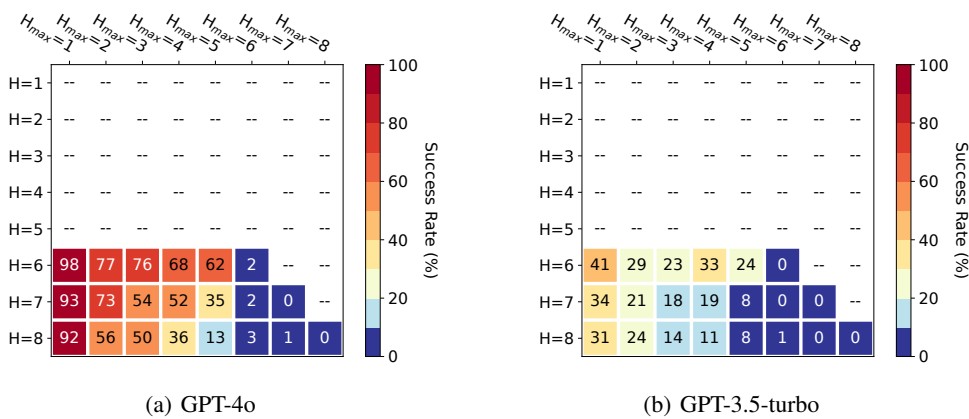

Figure 8: Success rate for predicting the sixth to last step.

## C.2 CANONICAL BOOLEAN FUNCTIONS

Examples of standard and CoT prompts for $(10, 4)$-parity and DNF with width 6 are given as follows:

- Standard prompt for parity:

```
Predict the output based on a pattern in the input binary string.

Input: 1010101100
Output: 0

Input: 0100000011
Output: 0

Input: 0110010000
Output: 1

Input: 1010100001
Output: 0

Input: 1010100001
Output: 0

...

Input: 1011000111
What is the output? Directly answer the question in the format
↪  'Output:'.
```

- CoT prompt for parity:

```
Replace some bits located at specific predefined positions in the
↪  binary string with 0 to form a new string. Then, based on some
↪  patterns in the new string to predict the output.

Input: 1000010001
New string: 0000010001
Output: 0

Input: 0100110111
New string: 0000110001
Output: 1

Input: 0101001000
New string: 0001000000
Output: 1

Input: 0100000010
New string: 0000000000
Output: 0

Input: 1011010000
New string: 0001010000
Output: 0

...

Input: 0100100110
What is the output? Directly answer the question in the format
↪  'New string:, Output:'.
```

- Standard prompt for DNF:

```
Predict the output based on a pattern in the input binary string.

Input: 001010 000001 010011
Output: 1

Input: 100111 011001 010111
Output: 1

Input: 010010 011010 101011
Output: 1

Input: 010101 001101 001001
Output: 1

Input: 111110 011001 010111
Output: 1

...

Input: 010001 110101 011011
What is the output? Directly answer the question in the format
↪  'Output:'.
```

- CoT prompt for DNF:

```
Replace some bits located at specific predefined positions in the
↪  binary string with 1 to form a new string. Then, based on some
↪  patterns in the new string to predict the output.

Input: 000101 011111 011010
New string: 100111 111111 011111
Output: 1

Input: 001101 100111 000101
New string: 101111 110111 000101
Output: 0

Input: 100001 011001 001010
New string: 100011 111011 001111
Output: 0

Input: 001100 010100 101011
New string: 101111 110110 101111
Output: 0

Input: 010101 111011 100101
New string: 110111 111011 100101
Output: 0

...

Input: 000100 011110 100000
What is the output? Directly answer the question in the format
↪  'New string:, Output:'.
```

