# OpenReview forum: "Chain-of-Thought Provably Enables Learning the (Otherwise) Unlearnable"
_ICLR.cc/2025/Conference — ICLR 2025 Poster_

### Official Review · Reviewer_PsRB · 2024-10-22

**Soundness:** 2
**Presentation:** 2
**Contribution:** 2
**Rating:** 6
**Confidence:** 3

**Summary:**

This paper investigates the impact of subtask complexity on the reasoning capabilities of Transformers when employing Chain-of-Thought (CoT). Theoretical results demonstrate the expressive power of Transformers to learn from in-context CoT samples through in-context gradient descent. Furthermore, the paper establishes upper and lower bounds for the error bound $\Delta(\mathcal{P}, h)$, highlighting the significance of task decomposition. Experimental results illustrate the effectiveness of CoT in parity and DNF tasks.

**Strengths:**

This work emphasizes the importance of understanding principles for designing CoT steps, providing several insights for creating more effective CoT formats when presenting in-context samples. The theoretical findings regarding error decomposition at each CoT intermediate step reveal the hardest subtask as a key bottleneck in reasoning.

**Weaknesses:**

1. While the paper proposes effective strategies for decomposing target problems as claimed in lines 23-25, these strategies appear limited to specific problems like parity or DNF. It remains uncertain whether these approaches can be generalized to CoT format design for a broader range of reasoning problems.

2. The theoretical results are somewhat unconvincing:
   - Lemma 1 indicates that a Transformer with depth linear to the number of reasoning steps $k$ and the number of gradient descent (GD) steps $t$ diverges from practical applications. Current large language models (LLMs) typically possess at most hundreds of layers, yet they can handle thousands of context lengths. Consequently, a construction involving constant-depth Transformers with polynomial hidden dimensions may yield more practical insights.
   - In Lemma 2, the bound becomes less meaningful when $k$ is not treated as a constant. However, reasoning lengths often are not constant; the expressive power is strictly constrained. [1] illustrates that a constant-depth, log-precision Transformer with polynomial hidden dimensions and either no CoT reasoning or $O(\log n)$ CoT reasoning steps cannot solve problems outside the $\mathsf{TC}^0$ class, where $n$ is the problem size. To strengthen the effectiveness of CoT, the reasoning steps $k$ should generally be $\Omega(\log n)$, and in many instances, $k = \text{poly}(n)$. Under these conditions, the bounds become quite loose.

[1] Chain of Thought Empowers Transformers to Solve Inherently Serial Problems, https://arxiv.org/abs/2402.12875

**Questions:**

1. Can you describe how to design effective CoT formats for more general problems using the strategies proposed in this paper?

2. What are the input and output formats (including CoT steps) for all experiments? What is the length of CoT reasoning steps for each task?

3. Lines 355-357 indicate that increasing reasoning steps $k$ negatively impacts performance. Can you elaborate on how this conclusion is drawn from the experiments in Section 5?

4. Is it feasible to develop a Transformer with constant depth as proposed in Lemma 1? For instance, could the filtering process mentioned in lines 303-305 be applied? A polynomial embedding size is acceptable in the context of constant layers, which aligns more closely with practical applications.

5. In lines 381-397, why is $|\mathcal{H}_1'|$ expected to be large when applying CoT, while $|\mathcal{H}_1'|=1$ without CoT? Given the problem setup, $\mathcal{A}$_CoT is an algorithm that learns from demonstrations, meaning the output predictor $h$ is influenced by the input demonstrations and is not necessarily unique. In this context, the assertion that $|\mathcal{H}_1'| = 1$ without CoT seems questionable.

---

> ### Author Response · Authors · 2024-11-20
>
> **Q1 & Weakness 1**
>
> As mentioned in Section 3.3, our general guidelines for forming an effective CoT are to decompose the hardest reasoning step into smaller steps that are easier to learn; parities and DNF serve as examples where we can provably show the task is unlearnable without CoT but learnable with CoT. While for more general problems it is typically infeasible to formally quantify the hardness, our guidelines still work by assessing the hardness of tasks in a qualitative or intuitive manner. For example, in the arithmetic reasoning task in Section 5.1, the hardness is qualitatively measured by the number of elementary functions required to construct the target function, and we found the performance is more significantly affected by the hardness of the hardest reasoning step in the CoT rather than the final task hardness.
>
> **Q2**
>
> The input and output formats of all experiments were provided in Appendix C.1 (for the arithmetic reasoning task in Section 5.1) and C.2 (for parities and DNF in Section 5.2). The CoT steps for the arithmetic reasoning task are $H-H_{max}+1$ since we hide $H_{max} -1$ steps in the original $H$ reasoning steps to control the hardness of the hardest step. The CoT steps for parities and DNF are $2$.
>
> **Q3**
>
> In Figure 2, as we increase $H$ (which is exactly the CoT steps when $H_{max} = 1$), the performance gradually drops. This is because, though each step is easy to learn, the error made at each step will accumulate and affect the final performance. This is consistent with our theory that increasing reasoning steps $k$ negatively impacts performance, despite the advantage of better expressiveness as pointed out in previous works such as [1].
>
> [1] Chain of Thought Empowers Transformers to Solve Inherently Serial Problems, https://arxiv.org/abs/2402.12875
>
> **Q4 & Weakness 2 (Lemma 1)**
>
> The filtering process mentioned in lines 303-305 can be applied to reduce the depth requirement from $kt$ to $t$; however, unless we treat $t$ as a constant, reducing the depth to a constant seems difficult. More generally, it is an open research question what learning algorithms Transformers can implement with only constant depth since most ICL works mentioned in the related work Section 1.2 also require at least $t$ depth. Nevertheless, all subsequent results after Section 3.2 / Lemma 1 are adaptable for other Transformer constructions. Namely, if future works identify more efficient ways for expressing the learning algorithm (say, with constant depth and polynomial embedding width), our results apply as well.
>
> **Q5**
>
> Note that in our setup, we do not consider learning only one target function $f$ (since being able to learn one function is not practically meaningful) but a family of target functions or a distribution family $\mathcal P = {(f_1, \mathcal D_1), (f_2, \mathcal D_2), \cdots }$. With CoT, the first-step learner is expected to output a different first-layer predictor $h_1$ for each unique first-layer target function $f_1$ in $\mathcal P_1$ (and of course for the same target function, the learner could also output a different predictor on random samples), and thus the possible number of first-step predictors $|\mathcal H_1’|$ is expected to be large in general. Without CoT, the learner also outputs a different predictor $h$ for each unique target function $f$ in $\mathcal P$; however, for each predictor (which outputs the prediction in just one step), it could be equivalently viewed as $h = h \circ \operatorname{Id}$ where the first step is the identity mapping and the second step is $h$ itself, and thus $|\mathcal H_1’| = 1$. Note that this is only for theoretical convenience for incorporating both cases in a single analytic framework; when CoT is not used, we do not actually provide intermediate steps in the demonstration or ask the learner to learn the identity mapping.
>
> **Weakness 2 (Lemma 2)**
>
> We treat $k$ as a constant since the error will accumulate and the upper bound does not hold asymptotically in $k$. This reveals a practical limitation of CoT when increasing the reasoning steps, which has not been revealed by previous theoretical works on CoT that purely focus on the expressiveness of Transformers. In experiments, we also find that when each step is equally easy to learn, increasing the reasoning steps often hurts the performance, which is consistent with our theoretical prediction.

---

> > ### Comment · Reviewer_PsRB · 2024-11-21
> >
> > Thanks for your clarification. However, I remain uncertain about several claims and results.
> >
> > **About Theorem 4 (Error Lower Bound) and Question 5**
> >
> > 1. > *"When CoT is used, we do not actually provide intermediate steps in the demonstration. "*
> >
> >    This statement appears contradictory. In Appendix C, the CoT prompting examples for increasingly complex functions (Appendix C.1, lines 1264–1289) clearly include intermediate steps in the demonstrations (e.g., steps 1–5 in lines 1271–1275 and lines 1278–1282). Similarly, the CoT prompts for parity (lines 1430–1457) and DNF (lines 1483–1509) provide "new strings" as intermediate steps. Could you clarify this apparent discrepancy?
> >
> > 2. For the size of $|\mathcal{H}_1'|$ when applying CoT, why can’t we equivalently represent $h_1$ as $h_1 \circ I$, where $I$ is the identity mapping? In other words, we could introduce an additional CoT step by letting the first step be the identity mapping, making $\mathcal{H}_1' =$ { $I$ }. Under this interpretation, the error lower bound would remain unchanged whether CoT is applied or not, which seems to undermine the argument for CoT's benefit. Could you address this point?
> >
> > **About the Relationship Between Performance and Reasoning Steps, and Question 3**
> >
> > 3. > *"In Figure 2, as we increase $H$ (which is exactly the CoT steps when $H_{max}=1$), the performance gradually drops. This is because, though each step is easy to learn, the error made at each step will accumulate and affect the final performance. This is consistent with our theory that increasing reasoning steps $k$ negatively impacts performance, despite the advantage of better expressiveness as pointed out in previous works such as [1]."*
> >
> >    > *[1] Chain of Thought Empowers Transformers to Solve Inherently Serial Problems, https://arxiv.org/abs/2402.12875*
> >
> >    This claim is not entirely convincing, given the experimental setup. The increase in $H$ correlates with an increase in the complexity of the target function. For example, focusing on the first column of Figure 2(a) or Figure 2(b), the increase in $H$ implies a more complex target function. Consequently, the observed drop in performance could be attributed to the increased difficulty of the tasks rather than the accumulation of reasoning errors. Furthermore, as stated in lines 483-485, the LLMs are provided with a fixed 10 demonstrations, regardless of the complexity of the target function. This limitation could also contribute to the performance drop. Thus, the evidence provided does not conclusively support the claim that increasing reasoning steps negatively impacts performance. Could you provide additional justification or experiments to isolate these variables?

---

> ### Author Response · Authors · 2024-11-21
>
> **About Theorem 4 (Error Lower Bound) and Question 5**
>
> 1. This is a typo, we meant "when CoT is NOT used, we do not actually provide intermediate steps in the demonstration or ask the learner to learn the identity mapping" in the previous response.
>
> 2. When applying CoT ($k=2$), each demonstration is given in the format $x, z, y$ and the predictor is $h_{k=2} = h_2 \circ h_1$ where both $h_1 \in \mathcal H_1$ and $h_2 \in \mathcal H_2$ are linear function classes on non-linear feature maps (per equation (7)); when CoT is not applied ($k=1$), demonstration is $x,y$ and the predictor is $h_{k=1} \in \mathcal H'$, which is a single linear function class on non-linear feature map (per equation (7)). The key point of theorem 4 is simply that $\mathcal H'$ is a more restricted function class than $\mathcal H_2 \circ \mathcal H_1$, and consequently there exists complex target functions that $\mathcal H_2 \circ \mathcal H_1$ can learn well while $\mathcal H'$ can not.  Neither writing $\mathcal H' = \mathcal H' \circ \{I\}$ or $\mathcal H_2 \circ \mathcal H_1 = \mathcal H_2 \circ \mathcal H_1 \circ \{I\}$ with change this result since they are essentially the same function class. However, theorem 4 only applies to two steps (note that theorem 4 states "for any decomposition operator T" rather than a sequence of decompositions $\{T_1, T_2, \ldots\}$) and does not deal with the composition of three function classes; in other words the remaining functions besides of the first step should be a linear function on non-linear feature map – both $\mathcal H' \circ \{I\}$ and $\mathcal H_2 \circ \mathcal H_1$ satisfy this while $ \mathcal H_2 \circ \mathcal H_1 \circ \{I\}$ do not.
>
> **About the Relationship Between Performance and Reasoning Steps, and Question 3**
>
> It is true for the same column in Figure 2 more reasoning steps implies more complex target functions, and our theoretical prediction from lemma 2 is that *if each step is equally easy to learn* the performance will hurt with more reasoning steps; in other words, it does not draw negative conclusions about the case where there are more reasoning steps but each step is easier to learn, since in this case each individual error $\Delta(P_i, A_i )$ could be smaller, causing tradeoffs between error accumulation and smaller individual error. In such a case where there are more reasoning steps but individual steps become easier (and the target function remains the same), our theoretical prediction from theorem 3 is that if the hardest step becomes easier, CoT would help despite error accumulation. This is empirically verified by checking the same row of Figure 2, i.e. the target function is the same, increasing $H_{max}$ (decreasing $k$) would significantly hurt the performance.

---

> > ### Comment · Reviewer_PsRB · 2024-11-22
> >
> > Thank you again for your response. However, I remain uncertain about the theoretical prediction derived from Lemma 2: "*if each step is equally easy to learn, the performance will hurt with more reasoning steps*".
> >
> > The observed performance drop seems to stem from task complexity rather than the number of reasoning steps. For instance, focusing on the first column of Figure 2(a), as the target function becomes more complex, solving the task inherently requires more reasoning steps, when the difficulty of each step is fixed. By Lemma 2, the error at each step accumulates, ultimately reducing performance. My argument is that task complexity is the primary driver of the performance drop, while the need for additional reasoning steps is a consequence rather than the root cause.
> >
> > I encourage the authors to clarify their claims (e.g., lines 367–369) regarding the relationship between performance and reasoning steps, especially in scenarios where task complexity increases.

---

> > > ### Author Response · Authors · 2024-11-22
> > >
> > > Thanks for the suggestion. You are correct that task complexity is an important factor of decreasing performance for the same column in figure 2; increasing the number of reasoning steps while ensuring each step is equally easy to learn naturally results in more complexity overall tasks in the experiment. If we fix the overall task hardness, increasing the reasoning steps will reduce the hardness of individual tasks and thus improve the performance as we have shown for the same row in figure 2. Both these phenomenon align with the theory.
> > >
> > > The descriptions under theorem 3 is not entirely clear and we have modified it to be more precise as the reviewer suggested. Please let us know if you have any other lingering concerns.

---

> > > > ### Comment · Reviewer_PsRB · 2024-11-24
> > > >
> > > > Thanks for your response and clarification. I'll increase my rating.

---

> > > > > ### Author Response · Authors · 2024-11-26
> > > > >
> > > > > Thanks for raising the score and for your support.

---

### Official Review · Reviewer_XoWP · 2024-11-02

**Soundness:** 3
**Presentation:** 3
**Contribution:** 3
**Rating:** 6
**Confidence:** 2

**Summary:**

This paper studies how chain-of-thought (CoT) impacts in-context learning (ICL) for transformer models. There are two main results. First, the paper shows that a linear-depth transformer model can implement a learning algorithm A_CoT that chains together k gradient descent learning algorithms A_i that run for t steps each for 1 <= i <= k (Lemma 1). An upper bound on error is given for A_CoT, thereby showing that a composite task is efficiently learnable by a transformer model if each subtask is efficiently learnable by each of the k individual gradient learning algorithms (Lemma 2), where the hardness of learning is bounded by the difficulty of the most difficult subtask (Theorem 3). Second, the authors lower-bound the error of A_CoT in the case of finite hypothesis class for A_1 as a function of the size of that hypothesis class (Theorem 4), which suggests that there may be functions that are learnable with CoT that are not learnable without CoT. A concrete witness is identified for learning parities (Corollary 5).

**Strengths:**

The paper studies an important problem in trying to theoretically elucidate CoT and ICL for transformer models. There is also preliminary experimental results that corroborate the theoretical results.

**Weaknesses:**

The presentation is quite dense could be improved to explain the motivation and intuitions behind definitions/results.
- For instance, the definition of in-context learning (Definition 1) makes a distinction between predictors and a learner (around line 195-196). From my understanding, the definition of Model(Data, query) = LearningAlg(Data)(query) = predictor_D(query) is just partial evaluation, and so the distinction that is drawn between predictors and learners is somewhat artificial. Rather, the main idea seems to me to formulate ICL, and so we just define a function that takes a query that is a tuple (demonstrations, x) as opposed to x. Moreover, it would be helpful to explain why PAC learning is chosen and to give pointers (presumably to Theorem 4) on how exactly PAC learning will be used to help you demonstrate your results and why it is sensible.
- Why does Definition 2 use multisets?
- Some of the material in the appendix on how your Transformers implement gradient descent result differs from Cheng et al. and Von Oswald et al.'s would be helpful in the main text to give more context.
- Could you offer more clarification on how to extend the results to handle k > 1 in lines 294-305 which is from what I understand is the main point of innovation of the paper. However, I can't seem to find more details in the main text or the appendix.
- The references for this paper do not use ICLR's reference style ...
- The statement of Lemma 1 might be helped by stating that t steps of gradient descent map to t layers of self-attention which is mentioned in the appendix.
- I find the statement of Theorem 4 to be somewhat strange. If I understood correctly, without CoT, we can essentially run for t gradient steps on a neural network that predicts from a representation that has t self-attention layers. With CoT for k steps, we can essentially run for kt gradient descent steps with kt self-attention layers and k prediction layers. How does Theorem 4 control for the same number of layers across reasoning steps k? Intuitively, I expect a theorem showing that there functions that are learnable with CoT that are not learnable without CoT to (1) show that a kt layer network run for kt gradient descent steps can not learn the task purely with an identity (trivial) task decomposition and (2) that a task decomposition (T in theorem 4) used by CoT can be constructed in polynomial time *uniformly* in the task with a Hypothesis class H_1'. However, there are no limits on the task decomposition T in the theorem and it's currently not clear to me how the size of the Transformer is controlled in the current statement of Theorem 4. Perhaps I misunderstood and would be curious to get more insight into what is actually going on.

**Questions:**

Please see weaknesses for questions.

---

> ### Author Response · Authors · 2024-11-20
>
> **Q1**
>
> It is standard in statistical learning to distinguish learner and predictor (e.g., see [1], Chapter 2.1): the learner is defined as a mapping from a set of samples to a function, i.e., $(\mathcal{X} \times \mathcal{Y})^N \rightarrow \mathcal{Y}^{\mathcal X}$, and the predictor (a.k.a. hypothesis) is defined as a function from $\mathcal X$ to $\mathcal Y$. In $\mathcal M_\theta(D,x) = \mathcal A(D)(x) = h_D(x)$, the first equality holds since mathematically any function $f(x_1, x_2)$ can be uniquely expressed in curried form as $g(x_1)(x_2)$, where $g$ is a function that maps an input $x_1$ to another function of $x_2$, i.e., $f(x_1, \cdot)$. The second equality is simply defining a notation (i.e., writing $g(x_1, x_2)$ as $h_{x_1}(x_2)$), which is also standard in learning theory.
>
> PAC learning is chosen since it provides a formal way to quantify the learnability of tasks in terms of the required number of samples (i.e., sample complexity) to successfully learn the task (i.e., achieve a desired level of accuracy and confidence). We show that with CoT, the sample complexity can be reduced to that of the hardest task, and without CoT, there exist tasks that are not PAC learnable regardless of how many samples are given and how many GD steps are used. These results formally quantify how CoT helps in learning initially complex tasks.
>
> [1] Understanding Machine Learning: From Theory to Algorithms
>
> **Q2**
>
> Definition 2 uses multisets since there might exist repeated elements. For instance, when the first target function is $f = f_2 \circ f_1$ and the second target function is $g = g_2 \circ f_1$, the subtask target function $f_1$ is repeated.
>
> **Q3**
>
> Thanks for the suggestion. We comprehensively discussed the connection in Appendix B and will find ways to incorporate some of them into the main text in the next version.
>
> **Q4**
>
> At a high level, to extend to $k > 1$, instead of using self-attention layers to simulate the training dynamics of the output $y$, as is done when $k = 1$, we construct self-attention layers to simulate the training dynamics of all intermediate results ${z_i}{i=1}^k$, including the output, as described in equations (20)-(23). To do so, we use a projection matrix to expand the dimension of embeddings from $d$ to $2d$, which allows it to store both the initial intermediate result $z_i$ and the prediction error (a.k.a. residual) $z_i - h_i(z{i-1})$. The former is used for computing the kernel for layer $i$ (i.e., the blue part in equation (9)), and the latter is used for the training dynamics of layer $i-1$ (i.e., the red part in equation (9)). $W_K, W_Q, W_V$ in equations (28) and (29) are constructed in such a way that they attend only to the part of embeddings that correspond to the current layer. The complete proof is given in Appendix A.1.
>
> **Q5**
>
> The formatting issue has been fixed.
>
> **Q6**
>
> We have updated the “Extension to $k > 1$” paragraph below Lemma 1 to make the connection between Transformer layers and GD steps clearer and more explicit.
>
> **Q7**
>
> Theorem 4 is an analysis of Algorithm 1, which holds for all numbers of GD steps $t$, all decompositions $T$, and compares $k=1$ (no CoT) and $k=2$ (one step of CoT). Particularly, the result in Theorem 4 is derived based on the representational limit of the linear hypothesis class used by the ICL algorithm. It states that regardless of how many GD steps are used and how many in-context samples are provided, the learning algorithm cannot successfully learn certain tasks (such as the parity example) without CoT, since the expected error always has a non-trivial lower bound. This forms a separation from using CoT, in which case we have shown that the task becomes learnable. With sufficient GD steps (which can be implemented by Transformers with a sufficient number of layers) and a sufficient number of in-context examples, the prediction error could be made arbitrarily small; this is otherwise impossible without CoT.
>
> Some other clarifications: The operator $T$ is predetermined and used to form the CoT examples (the learning algorithm itself does not construct CoT samples). Given a set of CoT examples generated based on the decomposition $T$, $\mathcal H’_1$ is the set of predictors the learning algorithm can return. In this case, without CoT, we could equivalently view the learning algorithm as always returning the identity mapping for the first step, i.e., $|\mathcal H’_1| = 1$.
>
> We will improve the presentation to make the results clearer, and please let us know if you have more questions.

---

> ### Comment · Reviewer_XoWP · 2024-11-21
>
> Thank you for your response.
>
> > Q1
>
> Thank you for the explanation. As I understand it now, you use the standard PAC learning definition with partial eval (curry/uncurry). My confusion stems from the phrasing "To define a notion of learnability in the context of language modeling, we **extend** the classic PAC learnability [42] to the setting in which the learner is associated with a parametric model" which made it seem like the standard definition was actually modified.
>
> > Q2
>
> Thank you for the response, this addresses my question.
>
> > Q3
>
> Thank you.
>
> >  Q4
>
> Thank you for this explanation. Perhaps it would help the reader if the "Extension to k > 1" header was placed before the introduction of Lemma 1, or Lemma 1 was modified to explicitly talk about the extension to k > 1.
>
> > Q5 and Q6
>
> Thank you.
>
> > Q7
>
> Thank you for the response. I still don't quite understand how the size of the transformer is controlled in the baseline (no CoT) vs. CoT case in theorem 4. Put another way, assuming that the baseline and CoT transformers have the same number of layers and are run for the same number of gradient steps (although they could be used differently), can you explain at a high-level why CoT overcomes these barriers? Is the "main insight" that the task decomposition (given to us by an oracle) means that we can target each reasoning step at a different loss function aligned with the appropriate task? I ask again because I believe this work addresses an important problem. However, I would like to understand in simple terms where all the heaving lifting is done, which as I understand it now, is given to you via the task decomposition oracle and the ability to choose a different loss function for each intermediate task.

---

> ### Author Response · Authors · 2024-11-21
>
> **Q1**
>
> We have modified the sentence to avoid ambiguity.
>
> **Q4**
>
> We have modified lemma 1 to explicitly state k>1.
>
> **Q7**
>
> In a high-level and informally, the results in section 4 state that: without CoT, the learning algorithm 1 (regardless of how many GD steps are used) is  unable to output predictors that can accurately solve the task $P$ if the task itself is inherently complex; in contrast, with CoT, instead of directly learning the task, the algorithm can learn two subtasks $P_1$ and $P_2$ that are easier than the original task $P$ and compose predictors obtained from each subtask to solve the initial task $P$. More specifically, directly learning the task is hard because the hypothesis class $\mathcal H$ is not sufficient (i.e. not expressive enough) to represent all target functions in $P$; decomposition makes learning easier because it enables learning $P$ step by step, that is we can first use $\mathcal H_1$ to learn $P_1$ and $\mathcal H_2$ to learn $P_2$ and then compose the learned predictors.
>
> It is important to note that theorem 4 itself does not directly link with Transformers — this is a result about algorithm 1. The way to link the results in section 4 to Transformers is by combining them with lemma 1, that is: without CoT, there does not exist such a Transformer (even it has infinite depth), under the construction we provided in lemma 1, that can learn the task; but with CoT, there indeed exists Transformers (with finite depth) that can simulate algorithm 1 and successfully learn the task. Overall, the insight here is not that “we can target each reasoning step at a different loss function aligned with the appropriate task”, but CoT makes learning a task easier by decomposing it into subtasks that are learnable by a learning algorithm (i.e. algorithms) Transformers can actually implement (otherwise without CoT more powerful algorithms are needed to learn the task but it’s an open question whether ICL can go beyond linear function classes considered in this paper).

---

> > ### Comment · Reviewer_XoWP · 2024-11-26
> >
> > Thank you for your explanation to Q7 which has clarified my misunderstanding.

---

> > > ### Author Response · Authors · 2024-11-26
> > >
> > > Thanks for your support!

---

### Official Review · Reviewer_dJXS · 2024-11-02

**Soundness:** 4
**Presentation:** 4
**Contribution:** 4
**Rating:** 10
**Confidence:** 4

**Summary:**

This paper analyzes Chain of Thought (CoT) learning in LLMs. The key idea behind CoT is to learn hard tasks by being shown intermediate steps involved in learning. The paper formulates this as in-context learning (ICL) which means that a parametric language model can be trained to generalize a set of demonstrations (with k steps) to predict the final output for a new input. The paper shows that CoT can be implemented by a step-by-step algorithm, where each step is generalized by a learning algorithm A_i from the examples of the i'th stage. The key lemma is to show that a transformer of depth linear in k can express the learning algorithm A_{CoT} = A_1...A_k as a sequence of gradient descent steps.
The main theorem then shows that CoT can efficiently learn a set of tasks which can be decomposed into subtasks with a sample complexity bounded by the sample complexity of the hardest subtask. Negatively, the paper gives a lower bound on the overall error as a function of hypothesis space and the problem class.
The concrete results show that the class of parity functions is not polynomially learnable in 1 stage, but can be learned in the CoT framework in 2 stages.
The paper introduces a new benchmark for CoT with controllable hardness. Each stage of each function manipulates two integers through addition and multiplication. The hardness is controlled by skipping some of the intermediate steps of the function. The success rates on GPT-40 and GPT-3.5 illustrate the predicted result - they degrade gradually with increasing the maximum number of contiguous missing steps.  They also show positive results on an (n,k)-parity task and on 3-term DNF learning using the same LLMs.

**Strengths:**

The paper gives a strong theoretical justification and analysis of CoT in the context of LLMs, building on prior work on understanding the prior work on viewing LLMs as few shot learners. The analysis is sound and well-presented.

The paper shows concretely how CoT makes otherwise unlearnable classes learnable through decomposition.

The empirical results support the theory and clarify the paper.

The paper presents some interesting benchmarks including a challenging task that can express arbitrary polynomials over integers, which can be controlled for hardness.

**Weaknesses:**

It appears that this work is somewhat related to `learning from exercises' in the problem solving setting (eg, https://dl.acm.org/doi/10.5555/93335.93344, and https://onlinelibrary.wiley.com/doi/10.1111/j.1467-8640.2008.00330.x) and curriculum learning (http://arxiv.org/abs/2101.10382). It would be good to see a discussion of the relationship.

**Questions:**

Since this work builds upon a number of previous works in CoT and transformers, please state precisely what was previously known and what are the novel (theoretical) contributions.

It is not always obvious how to decompose a given task. Do you have any advice on how to approach this on a practical problem?

The importance of the work is diminished by having to manually decompose each new task to generate demonstrations. Do you have any thoughts on how to reduce this burden?

---

> ### Author Response · Authors · 2024-11-20
>
> **Weakness**
>
> Thank you for the suggestion. The analysis of CoT in this paper is indeed conceptually related to the idea of learning from easier to more difficult tasks. We will discuss the connection in the related work section.
>
> **Q1**
>
> As discussed in Section 1.2, previous theoretical works on CoT either focused on overly restricted tasks or did not quantify the effects of different ways to decompose the task. This paper formalizes in-context learnability and first explores how different ways of task decomposition affect learnability.
>
> **Q2**
>
> Our results suggest that to improve the overall learnability of a complex task, it is desirable to decompose the hardest step into smaller steps, where the hardness here is quantified by the sample complexity. Unfortunately, in practice, except for certain well-defined problems such as parities and DNF considered in this work, there lacks a unified way to quantify the hardness, and the specific way to decompose the task should be studied on a case-by-case basis (e.g., in our experiments, learning higher-order polynomials is generally considered harder).
>
> **Q3**
>
> One potential way to eliminate manual efforts for designing CoT is to automate this process by using ‘find the hardest step and decompose it into subproblems’ or something similar as a prompt and instruct the LLMs to construct demonstrations under our principle, following existing works such as zero-shot CoT or auto CoT [1,2].
>
> [1] Large language models are zero-shot reasoners, NeurIPS 2022
>
> [2] Automatic chain of thought prompting in large language models, ICLR 2023

---

### Official Review · Reviewer_S1ny · 2024-11-03

**Soundness:** 3
**Presentation:** 3
**Contribution:** 3
**Rating:** 6
**Confidence:** 3

**Summary:**

This paper studies the effectiveness of chain of thought (CoT) in the in-context learning setting. Particularly, it is shown that there is a parametrization of the Transformer architecture that allows in-context learning of certain functions with CoT. It is further claimed that there are functions that are only learnable with CoT and not otherwise in this setting. The paper concludes with experiments with GPT models and their in-context learning abilities based on the granularity of the CoT provided to them.

**Strengths:**

- The paper is generally well-written and theoretically studies an important problem: ‘effectiveness of CoT in the in-context learning setting’. Moreover, the paper generally proposes rather clear definitions of in-context learning with and without CoT based on the PAC framework.
- A particularly interesting theoretical prediction is that making CoT more granular and longer does not necessarily improve the performance.

**Weaknesses:**

- One problem with in-context learning works as done in this paper is that it is only proven that there is a Transformer parametrization suitable for in-context learning (similar to an expressivity result). However, it is not clear whether this parameterization is realistic or not. I.e., if a Transformer pre-training is done in a reasonable manner, it’s not clear if it's possible/likely that the Transformer achieves the needed parameterization. (E.g., the claimed parametrization may not be reachable by GD in practice and there is generally no insight on the pre-training process.)

**Questions:**

- The depth requirement in Theorem 1, $kt$, seems rather impractical and quite large. Further, I think there are several clarifications required. For start, I think $t$ itself is not a parameter of learning setting but part of the learning algorithm. We are in the PAC setting in this paper. So it would indeed make sense to give the depth of Transformer based on $k, \delta, \epsilon$ instead. In particular, for me it’s not clear if the complexity in $k$ is really linear. Based on eq. (11) we know that as $k$ increases, we have a compounding and increasing error. Thus, probably more GD iterations are also needed. I.e., for a given $\epsilon, \delta$, parameter $t$ is not proven to be independent of $k$. So effectively, I’m not sure if the depth is linear in $k$ as claimed. I think giving the depth in $k, \delta, \epsilon$ would fix all these issues.
- The negative result for the in-context learning is given for the hypothesis class in eq. (7) and not Transformers. Can we say something about Transformers not being able to learn the functions in question in an in-context fashion?
- In definition 1, part 2, how is $\mathbb{E}_x[l(h(x),f(x))] \leq \epsilon$ with prob. $1-\delta$ equivalent with $\Delta(\mathcal{P}, h) \leq \epsilon$?
- For Theorem 3, it is said that the number of samples we need is equal to the number of samples needed for the hardest subtask. However, the samples used by different subtasks are not independent. Can you further clarify how is this taken into account?
- Given a fixed architecture (with fixed depth), can you please further elaborate on the effect of making CoT more granular and also the effect of increasing the number of in-context samples?
- In the in-context setting, when we go the in-context setting we provide more info by giving each sample. In other words, comparing the sample complexities with and without CoT doesn’t seem to be totally fair. Maybe, can you give a comparison based on the total number of tokens given?



Suggestions/Minor remarks:
- For me, the notation $\Delta(\mathcal{P}, h)$ used in equation 5 is not clear. $h$ on the right hand side depends on the function being learned however the $h$ is on the left hand side should be independent of it and represent the learning algorithm. Maybe replace it with $\mathcal{A}$?
- I would suggest that the authors include the existence of a hypothesis class in page 2 to increase the clarity.

---

> ### Author Response · Authors · 2024-11-20
>
> **Q1**
>
> The number of the required steps $t$ might indeed increase with $k$ given fixed $\epsilon$ and $\delta$ due to compounding errors, and thus the depth dependence on $k$ might not be linear (if the goal is to achieve a fixed small error). While we could explicitly account for the dependency by, e.g., writing $t$ in terms of $\epsilon$ and $\delta$, it introduces additional technical difficulties since their relations depend also on the learning task, which is not specified in both theory and practice. For some tasks, such as parities in Section 4.2, it is infeasible to write $t$ in terms of $\epsilon$ and $\delta$ since the error is lower bounded and we could not achieve small error regardless of how large $t$ is. Therefore, in this paper, we do not restrict the depth of the Transformer and allow $t$ to be chosen arbitrarily (i.e. one that is optimal for the task). While in practice the depth of Transformers is restricted (which is a limitation shared by most in-context learning works), we nonetheless have made real-world predictions on how CoT affects the in-context learning performance, verified by our experiments.
>
> **Q2**
>
> It is prohibitively hard (and an open research question) to discuss in general what tasks Transformers cannot learn, without fixing a specific (or a class of) learning algorithms, as we do in this work. While we have thought about deriving information-theoretic lower bounds for certain tasks to show Transformers cannot in-context learn these tasks, we have not found separation results showing how CoT could help in this case. We hope to cover this in future work.
>
> **Q3**
>
> This is a typo. We meant $\forall (f, \mathcal{D}) \in \mathcal{P}, \mathbb{E}_{x \sim \mathcal{D}(x)}[l(h(x), f(x))] \leq \epsilon$ is equivalent to $\Delta(\mathcal{P}, h) \leq \epsilon$, and you are correct they are not equivalent with an additional probability constraint. Since all theoretical results are derived based on $\Delta(\mathcal{P}, h)$, we modified Definition 1 accordingly to avoid ambiguity.
>
> **Q4**
>
> It does not matter whether or not the samples for different subtasks are independent, since the probability (w.r.t. a random draw of samples) that the learner fails to learn the overall task is upper bounded by the probability that each individual subtask fails, even if a sample from one subtask is dependent on a sample from another task (i.e., by Bonferroni inequality (44)). Therefore, to successfully learn the overall task, it suffices to choose the sample number based on the hardest subtask. See the complete proof in Appendix A.3.
>
> **Q5**
>
> For a fixed architecture and assuming the depth is sufficiently large to express the learning algorithm where $t$ is set to be optimal (which is considered in our analysis), increasing the number of CoT steps would reduce the sample complexity to that of the hardest subtask. However, suppose the hardest subtask of two different CoTs has the same sample complexity; the one with the longer chain would suffer from the compounding error issue and thus overall perform worse. Increasing the number of in-context examples is always beneficial in our theory. These results have been verified in our experiments. For the case where the depth is insufficient to express the learning algorithm where $t$ is chosen to be optimal (which we have not discussed), there will exist more complicated tradeoffs between $k$ and $t$, as you mentioned, and these are generally prohibitive to quantify (which seems to be an open research question even beyond the scope of ICL/CoT/LLM).
>
> **Q6**
>
> If we restrict the number of total tokens, there will be a tradeoff between the number of samples and the number of CoT steps, which is generally hard to quantify if we do not consider a specific task and a specific form of CoT. However, for the parity example discussed in Section 4.2, since without CoT the lower bound in Equation (12) holds regardless of the sample size (and the number of tokens), whereas with CoT the problem becomes learnable, CoT provably has benefits for this particular example.
>
> **Minors**
>
> Thank you for the suggestion. We have modified the notation to $\Delta(\mathcal{P}, \mathcal{A})$ and will revise the introduction accordingly.

---

> > ### Comment · Reviewer_S1ny · 2024-11-30
> >
> > Thank you for the answers. I have accordingly increased my score. I would suggest the authors include the discussion on the trade-offs and relations between $t, k, \delta, \epsilon$  in the revised versions of the paper.

---

> > > ### Author Response · Authors · 2024-12-01
> > >
> > > Thank you for raising the score, we will include some of the discussions in the next version of the paper.

---

### Author Response · Authors · 2024-11-26

Dear Reviewer S1ny, dJXS, XoWP,

Thanks for your time reviewing this paper and invaluable suggestions for improvement.

As we are nearing the end of the discussion, we would greatly appreciate any feedbacks on whether our responses have addressed your questions or concerns in the initial reviews. If there are any remaining issues or further questions, we would be happy to address them.

Thanks,
Authors

---

### Meta-Review · Area_Chair_SvYz · 2024-12-18

**Metareview:**

The reviewers were generally positive about the paper, specifically about how well the paper has been written, the theoretical justifications provided and the empirical results presented in the paper. We hope that the authors will take reviewers' feedback (especially about the missing references that can be better related to, the assumptions for the theorems and lemmas that the authors have promised in the rebuttals, and expanding on the experimental setup and details, another promise from the authors in the rebuttal) and improve the paper for the final version.

**Additional Comments On Reviewer Discussion:**

There was some sort of consensus towards acceptance.

---

### Decision · Program_Chairs · 2025-01-22

Accept (Poster)